# A Linear Speedup Analysis of Distributed Deep Learning with Sparse and Quantized Communication

**Peng Jiang**
The Ohio State University
jiang.952@osu.edu

**Gagan Agrawal**
The Ohio State University
agrawal@cse.ohio-state.edu

## Abstract

The large communication overhead has imposed a bottleneck on the performance of distributed Stochastic Gradient Descent (SGD) for training deep neural networks. Previous works have demonstrated the potential of using gradient sparsification and quantization to reduce the communication cost. However, there is still a lack of understanding about how sparse and quantized communication affects the convergence rate of the training algorithm. In this paper, we study the convergence rate of distributed SGD for non-convex optimization with two communication reducing strategies: sparse parameter averaging and gradient quantization. We show that $\mathcal{O}(1/\sqrt{MK})$ convergence rate can be achieved if the sparsification and quantization hyperparameters are configured properly. We also propose a strategy called periodic quantized averaging (PQASGD) that further reduces the communication cost while preserving the $\mathcal{O}(1/\sqrt{MK})$ convergence rate. Our evaluation validates our theoretical results and shows that our PQASGD can converge as fast as full-communication SGD with only $3\% - 5\%$ communication data size.

## 1   Introduction

The explosion of data and an increase in model size has led to great interest in training deep neural networks on distributed systems. In particular, distributed *stochastic gradient descent* has been extensively studied in both deep learning and high-performance computing communities, with the goal of accelerating large-scale learning tasks [13, 10, 9, 47, 3, 14, 30, 31, 6, 44, 38]. In today's mainstream deep learning frameworks such as Tensorflow, Torch, MXNet, Caffe, and CNTK [22, 1, 11, 8, 36], *data-parallel* distributed SGD is widely adopted to exploit the compute capacity of multiple machines.

The idea of data-parallel distributed SGD is that each machine holds a copy of the entire model and computes stochastic gradients with local mini-batches, and the local model parameters or gradients are frequently synchronized to achieve a global consensus of the learned model. In this context, a well-known performance bottleneck is the high bandwidth cost for synchronizing the gradients or model parameters among multiple machines [3, 27, 20, 28, 9, 42, 29]. A popular approach to overcome such a bottleneck is to perform *compression* of the gradients [33, 4, 46, 5, 37]. For example, Aji *et al.* [4] propose to *sparsify* the gradients and transmit only the components with absolute values larger than a threshold. Their sparsification method reduces the gradient exchange and achieve 22% speedup gain on 4 GPUs for a neural machine translation task. Wen *et al.* [46] propose to aggressively *quantize* the gradients to three numerical levels {-1, 0, 1}. Their quantization method reduces the communication cost with none or little accuracy lost on image classification.

Though numerous variants of gradient quantization and/or sparsification have been proposed and successfully applied to different deep learning tasks [42, 45, 37, 48, 18, 7, 16], their impact to the convergence rate of distributed SGD (especially for *non-convex* optimization) is still unclear. Most of the research efforts involve a simple empirical demonstration of convergence. Wen *et al.* [46]

show the convergence of their ternary gradient method with a strong assumption on the gradient bound, and yet no convergence rate is given. Alistarh *et al.* [5] analyze the convergence rate of SGD with quantized gradients – however, their result on non-convex optimization shows that the "variance blowup" caused by quantization is constant. Overall, there is not a good understanding of how gradient sparsification and quantization impact the convergence rate of distributed SGD, whether they are worth applying in general, and how they can be applied properly to achieve the optimal convergence rate and performance.

To fill these gaps in theory, this paper studies the convergence rate of distributed SGD with sparse and quantized communication for the following non-convex stochastic optimization problem:

$$\min_{x \in \mathbb{R}^N} f(x) := \mathbb{E}_{\xi \sim \mathcal{D}} F(x; \xi), \tag{1}$$

where $x \in \mathbb{R}^N$ are the model variables, $\mathcal{D}$ is a predefined distribution and $\xi$ is a random variable referring to a data sample, both $F(x; \xi)$ and $f(x)$ are smooth (but not necessarily convex) functions. This formulation summarizes many popular machine learning models including deep learning [25].

**Contributions** We first analyze the convergence rate of distributed SGD with two communication reducing techniques: sparse parameter averaging and gradient quantization. For sparse parameter averaging, we prove that distributed SGD can maintain its asymptotic $\mathcal{O}(1/\sqrt{MK})$ convergence rate as long as we can make sure all of the parameter components are exchanged in a limited number of consecutive iterations. Here, $M$ is the total mini-batch size on $n$ nodes, and $K$ is the number of iterations. As a corollary, we prove that distributed SGD that averages the model parameters only once every $p$ iterations can converge at rate $\mathcal{O}(1/\sqrt{MK})$.

For gradient quantization, we prove that if using *unbiased stochastic* quantization function, distributed SGD will converge at rate $\mathcal{O}((1 + q)/\sqrt{MK})$ or $\mathcal{O}((1 + qm)/\sqrt{MK})$, depending on if the training data are shared or partitioned among nodes. Here, $m$ is the mini-batch size on a single node, and $q$ is a bound of the expected quantization error we define later. This result suggests that choosing a quantization function that ensures $q = \Theta(1)$ and $q = \Theta(1/m)$ can achieve $\mathcal{O}(1/\sqrt{MK})$ convergence rate for distributed SGD in the two scenarios.

The $\mathcal{O}(1/\sqrt{MK})$ convergence rate is usually desired in distributed training as it implies linear speedup across multiple machines w.r.t computation complexity [14, 32]. Our analysis results indicate that distributed SGD with sparse and quantized communication can converge as fast as full-precision SGD if configured properly. Intuitively, the $\mathcal{O}(1/\sqrt{MK})$ convergence rate can be preserved because the additional deviation of the gradient introduced by sparsification or quantization can be relatively small compared with the deviation caused by the *stochastic* method itself.

According to the analysis results, to ensure the optimal convergence rate, sparsification or quantization alone achieves limited compression ratio. To further reduce the communication cost without impairing the convergence rate, we propose to communicate *quantized* changes of model parameters once every $p$ iterations. We prove that our algorithm converges at rate $\mathcal{O}((1 + q)/\sqrt{MK})$ if the training data are shared among all nodes, and converges at rate $\mathcal{O}((1 + mpq)/\sqrt{MK})$ if the training data are partitioned. By properly setting $p$ and $q$, we achieve $\mathcal{O}(1/\sqrt{MK})$ convergence rate for our algorithm.

## 2  Related Work

Many gradient sparsification and quantization techniques have been proposed to reduce the communication overhead in distributed training.

**Gradient Sparsification** Strom [42] proposed to only send gradient components larger than a predefined threshold – however, the threshold is hard to determine in practice. Aji *et al.* [4] presented a heuristic approach to truncate the smallest gradient components and only communicate the remaining large ones. They saved 99% of gradient exchange with 0.3% loss of BLEU score on a machine translation task. Lin *et al.* [33] showed that techniques such as momentum correction and local gradient clipping can help the convergence of distributed SGD with sparse gradient exchange. They achieved a gradient compression ratio from 270x to 600x for a wide range CNNs and RNNs without losing accuracy. Despite the good performance in practice, the gradient sparsification methods in previous works are largely heuristic and no convergence guarantee has been established.

**Gradient Quantization** Seide *et al.* [37] proposed to use only 1-bit to represent the gradient. They achieve 10x speedup for a speech application. Alistarh *et al.* [5] proposed a quantization method named QSGD and gave its convergence rate for both convex and non-convex optimization; however, their convergence bound for the non-convex case has a constant term that does not converge over iterations. Wen *et al.* [46] proposed to aggressively quantize the gradients to three levels {-1,0,1}. They proved that their algorithm converges *almost surely* under a strong assumption on the gradient bound; however, no convergence rate is given. Moreover, none of the previous works have considered the impact of partitioned training data to the convergence rate of distributed SGD with gradient quantization.

There are also efforts that quantize the entire model including the gradients. For example, Buckwild! [12] showed the convergence guarantee of low-precision SGD with assumptions on convexity and gradient sparsity. Li *et al.* [26] studied different training methods for quantized neural network. They prove that convergence is guaranteed for training quantized neural network with convexity assumption. They also explained the inherent difficulty in training quantized neural network for non-convex cases. In this work, we focus on full precision neural network with quantized gradients, and we show that convergence rate is guaranteed on non-convex optimization if a good quantization function is used.

## 3   Analysis of Sparse Parameter Averaging and Gradient Quantization

In this section, we analyze the convergence rates of distributed SGD with two communication reducing techniques: sparse parameter averaging and gradient quantization.

### 3.1   Notation and Assumptions

We focus on *synchronous data-parallel* distributed SGD. The original objective defined in (1) can be rewritten as:

$$\min_{x \in \mathbb{R}^N} f(x) := \frac{1}{n} \sum_{i=1}^{n} \underbrace{\mathbb{E}_{\xi \sim \mathcal{D}_i} F_i(x; \xi)}_{=: f_i(x)}, \tag{2}$$

where $\mathcal{D}_i$ is a predefined distribution on the $i$th node. If the training data are shared among all nodes, then $\mathcal{D}_i$'s are the same as $\mathcal{D}$. If data are partitioned and placed on different nodes and each node defines a distribution for sampling local data, then $\mathcal{D}_i$'s are different.

**Notation**

- $\|\cdot\|_2$ denotes the $\ell_2$ norm of a vector or the spectral norm of a matrix.
- $\|v\|_1 := \sum_i |v_i|$ denotes the $\ell_1$ norm of a vector.
- $\|v\|_\infty := \max_i |v_i|$ denotes the maximum norm of a vector.
- $\|\cdot\|_F$ denotes the Frobenius norm of a matrix.
- $\nabla f(\cdot)$ denotes the gradient of a function $f$.
- $\mathbf{1}_n$ denotes the column vector in $\mathbb{R}^n$ with 1 for all elements.
- $\mathbf{e}_i$ denotes the column vector in $\mathbb{R}^n$ with 1 for the $i$th element and 0 for others.
- $f^*$ denotes the optimal solution of (2).

**Assumptions**

- All $\nabla f_i(\cdot)$'s are Lipschitz continuous with respect to the $\ell_2$ norm, that is,
$$\left\| \nabla f_i(x) - \nabla f_i(y) \right\|_2 \le L \|x - y\|_2 \quad \forall x, \forall y, \forall i. \tag{3}$$
- The stochastic gradient $\nabla F_i(x; \xi)$ is unbiased, that is,
$$\mathbb{E}_{\xi \sim \mathcal{D}_i} \nabla F_i(x; \xi) = \nabla f_i(x) \quad \forall x. \tag{4}$$
- The variance of stochastic gradient is bounded, that is,
$$\mathbb{E}_{\xi \sim \mathcal{D}_i} \left\| \nabla F_i(x; \xi) - \nabla f_i(x) \right\|_2^2 \le \sigma^2 \quad \forall x, \forall i. \tag{5}$$

---

**Algorithm 1** The procedure on the $i$th node of distributed SGD with sparse gradients

---

**Require:** initial point $x_{0,i}$, number of iterations $K$, and learning rate $\gamma$
1: **for** $j = 0, 1, 2, \ldots, K - 1$ **do**
2:     Randomly select $m$ training samples indexed by $\xi_{j,i} = [\xi_{j,i,0}, \xi_{j,i,1}, \ldots, \xi_{j,i,m-1}]$
3:     Compute a local stochastic gradient based on $\xi_{j,i}$: $\nabla F_i(x_{j,i}; \xi_{j,i})$
4:     Update the model parameters locally: $x_{j+\frac{1}{2},i} = x_{j,i} - \gamma \nabla F_i(x_{j,i}; \xi_{j,i})$
5:     Select a subset of parameter components indexed by $v_j$ and let $\mathcal{P}_j = v_j v_j^T$
6:     Average the selected parameter components: $x_{j+1,i} = \frac{1}{n} \sum_{k=1}^{n} \mathcal{P}_j x_{j+\frac{1}{2},k} + (\mathcal{I} - \mathcal{P}_j) x_{j+\frac{1}{2},i}$
7: **end for**

---

- The variance of gradient among nodes is bounded, that is,
$$\mathbb{E}_{i \sim \mathcal{U}(1,n)} \big\| \nabla f_i(x) - \nabla f(x) \big\|_2^2 \leq \varsigma^2 \quad \forall x, \tag{6}$$
  where $\mathcal{U}(1, n)$ is a discrete uniform distribution of integers from 1 to $n$. If all nodes share the same training data, $\varsigma = 0$.

These assumptions are commonly used in previous works for analyzing convergence rate of distributed SGD [46, 31, 5, 17].

### 3.2 Sparse Parameter Averaging

Sparse parameter averaging aims to reduce the communication overhead by exchanging only a subset of gradient or model parameter components in each iteration. Algorithm 1 gives a procedure of distributed SGD with sparse parameter averaging. All nodes are initialized with an identical initial point $x_0$. In each iteration, it first computes a stochastic gradient based on the current value of model parameters and a local mini-batch. Then, the model parameters are updated locally with the stochastic gradients. Next, a subset of parameter components are selected (Line 5). The selected components are denoted as a binary vector $v_j$ in the algorithm – the $p$th element in $v_j$ is 1 if the $p$th component is selected; otherwise it is 0. We require that all nodes share the same $v_j$ in the $j$th iteration, i.e., all nodes are communicating the same components. This is a reasonable requirement as averaging different components from different nodes leads to inconsistent model parameters. The selection is applied to the model parameters via a projection $\mathcal{P}_j = v_j v_j^T$ which is a diagonal matrix with the $p$th element in the diagonal being 1 if the $p$th component is selected. In the synchronization step (Line 6), the selected components are updated with the average values from all nodes while the unselected components keep their local values. Note that Algorithm 1 is different from *asynchronous* SGD [35] as all nodes are synchronized before updating the selected local model parameters in this algorithm.

**Convergence Rate** Throughout the paper, we discuss the convergence rate of training algorithms in terms of the average of the $\ell_2$ norm of the gradients. Specifically, we say an algorithm gives an $\epsilon$-approximation solution if
$$\frac{1}{K} \left( \sum_{j=0}^{K-1} \big\| \nabla f(x_j) \big\|_2^2 \right) \leq \epsilon. \tag{7}$$

This metric is conventionally used in the analysis for non-convex optimization [17, 32, 31]. Instead of considering any specific sparsification strategy, we require Algorithm 1 to exchange all parameter components in any $p$ consecutive iterations. More formally, we require
$$\prod_{t=j}^{k} (\mathcal{I} - \mathcal{P}_j) = 0 \qquad \forall (k-j) \geq p. \tag{8}$$

If this condition holds, we have the following convergence rate for distributed SGD with any gradient sparsification method:

**Theorem 1.** *Under the assumptions in §3.1, if $L\gamma \leq 1$ and $1 > 6np^2 L^2 \gamma^2$, Algorithm 1 has the following convergence rate:*

$$\frac{1}{K} \left( \sum_{j=0}^{K-1} \left\| \nabla f\left( \frac{X_j \mathbf{1_n}}{n} \right) \right\|_2^2 \right) \leq \frac{2\left(f(x_0) - f^*\right) D_1}{\gamma K} + \frac{L\gamma \sigma^2 D_1}{M} + \frac{pn^2 L^2 \gamma^2 \sigma^2 D_2}{M} + 3np^2 L^2 \gamma^2 \varsigma^2 D_2,$$

$$\tag{9}$$

*where*

$$D_1 = \left( \frac{1 - 3np^2 L^2 \gamma^2}{1 - 6np^2 L^2 \gamma^2} \right) , \quad D_2 = \left( \frac{1}{1 - 6np^2 L^2 \gamma^2} \right). \tag{10}$$

In Theorem 1, $X_j = [x_{j,1}, x_{j,2}, \ldots, x_{j,n}] \in \mathbb{R}^{N \times n}$ are the model parameters on $n$ nodes and $\frac{X_j \mathbf{1_n}}{n} = \frac{1}{n} \sum_{i=1}^{n} x_{j,i}$ is their average. Theorem 1 indicates that Algorithm 1 has guaranteed convergence if $L\gamma \leq 1$ and $1 > 6np^2 L^2 \gamma^2$. The result applies to any sparsification strategy as long as a limited $p$ is ensured. Our analysis shows that the threshold used in different sparsification strategies for selecting gradient components actually does not affect the convergence rate of SGD directly; instead, it is *the variance of model parameters among different computing nodes* that really matters. (See Proof to Theorem 1 in supplemental for more details.) Intuitively, a larger threshold means fewer exchanges of gradient components, which leads to a larger variance of model parameters among nodes, and thus, a slower convergence of the training process. The analysis also provides a theoretical basis for tuning the threshold for selecting gradient components in sparse-communication SGD: we should adjust the threshold adaptively to keep a small variance of model parameters among nodes.

With a proper learning rate, we can obtain the following result from Theorem 1:

**Corollary 1.** *Under the assumptions in §3.1, if setting $\gamma = \theta \sqrt{M/K}$ where $\theta > 0$ is a constant, we have the convergence rate for Algorithm 1 as:*

$$\frac{1}{K} \left( \sum_{j=0}^{K-1} \left\| \nabla f \left( \frac{X_j \mathbf{1_n}}{n} \right) \right\|_2^2 \right) \leq \frac{4\theta^{-1} \left( f(x_0) - f^* \right) + 2\theta L\sigma^2}{\sqrt{MK}} + \frac{2pn^2\theta^2 L^2 \sigma^2 + 6nM\theta^2 p^2 L^2 \varsigma^2}{K} \tag{11}$$

*if the total number of iterations is large enough:*

$$K \geq 12nM\theta^2 p^2 L^2. \tag{12}$$

If $K$ is large enough, the second term in (11) will be dominated by the first term, and the algorithm will converge at rate $\mathcal{O}(1/\sqrt{MK})$. In practice, $K$ is usually set as a fixed number. Corollary 1 indicates that there is a trade-off between the communication overhead and the convergence rate. A larger $p$ leads to a smaller communication overhead, but a larger second term in (11).

**Linear Speedup** With $\mathcal{O}(1/\sqrt{MK})$ convergence rate, Algorithm 1 achieves $\epsilon$-approximation solution when $mK \propto 1/(n\epsilon^2)$. That is, the amount of computation required on each node is inversely proportional to $n$. Thus, linear speedup can be achieved by Algorithm 1 asymptotically w.r.t computational complexity.

**Periodic Averaging** As a particular sparsification strategy, we can average the model parameters once every $p$ iterations. That is, $\mathcal{P}_j = \mathcal{I}$ if $p$ divides $j$, and $\mathcal{P}_j = \mathbf{0}$ if $p$ does not divide $j$. This strategy obviously satisfies the condition in (8). Therefore, we have the following result:

**Corollary 2.** *If we periodically average the model parameters once every $p$ iterations, distributed SGD still converges at rate $\mathcal{O}(1/\sqrt{MK})$ when $K$ is large enough.*

One advantage of this simple strategy is that it not only decreases the average communication data size but also reduces the average latency by a factor of $p$. This approach has been used in practice [34, 43]; however, its convergence has not been well studied. Our result illustrates the interaction of $p$ with other training hyperparameters and shows its influence to the convergence rate of distributed SGD.

## 3.3 Gradient Quantization

As an orthogonal approach to sparse parameter averaging, gradient quantization aims to reduce the communication cost by representing the gradients with fewer bits. The local stochastic gradient on each node is given to a quantization function before synchronizing with other nodes. The model parameters are then updated with the average of the quantized gradients.

**Convergence Rate** Previous works have shown the convergence of distributed SGD with *unbiased stochastic* quantization functions [5, 46]. However, no convergence rate is given in [46], and the convergence analysis on non-convex optimization in [5] shows that the average $\ell_2$ norm of the gradients has a constant *variance blowup* term. We now give a more general convergence result for

distributed SGD using unbiased stochastic quantization functions. Suppose the quantization function is $Q$, our result is based on the bound of expected error of a quantization function defined as follows:

$$q = \sup_{x \in \mathbb{R}^N} \frac{\left\| Q(x) - x \right\|_2^2}{\left\| x \right\|_2^2}. \tag{13}$$

With this definition, we have the following convergence result for distributed SGD with gradient quantization:

**Theorem 2.** *Under the assumptions in §3.1, if using an unbiased quantization function with an error bound $q$ as defined in (13) and $(1 + \frac{q}{n})L\gamma < 2$, distributed SGD has the following convergence rate:*

$$\frac{1}{K} \left( \sum_{j=0}^{K-1} \left\| \nabla f\left( x_j \right) \right\|_2^2 \right) \leq \frac{(f(x_0) - f^*)D}{\gamma K} + \frac{(1+q)L\gamma\sigma^2 D}{2M} + \frac{qL\gamma\varsigma^2 D}{2n}, \tag{14}$$

*where*

$$D = \frac{2}{2 - (1 + \frac{q}{n})L\gamma} \tag{15}$$

Setting a proper learning rate, we can obtain the following result:

**Corollary 3.** *Under the assumptions in §3.1, if using a quantization function with an error bound of $q$ and setting $\gamma = \theta\sqrt{M/K}$ where $\theta > 0$ is a constant, we have the following convergence rate for distributed SGD:*

$$\frac{1}{K} \left( \sum_{j=0}^{K-1} \left\| \nabla f\left( x_j \right) \right\|_2^2 \right) \leq \frac{2\theta^{-1}\left( f(x_0) - f^* \right) + (1+q)\theta L\sigma^2}{\sqrt{MK}} + \frac{m}{\sqrt{MK}}\theta q L\varsigma^2 \tag{16}$$

*if the total number of iterations is large enough:*

$$K \geq ML^2\theta^2(1 + \frac{q}{n})^2. \tag{17}$$

Corollary 3 suggests that if all nodes share the same training data (i.e. $\varsigma = 0$), $q = \Theta(1)$ is a sufficient condition for distributed SGD to achieve $\mathcal{O}(1/\sqrt{MK})$ convergence rate. If each node can access only a partition of the training data (i.e. $\varsigma \neq 0$), distributed SGD can still achieve $\mathcal{O}(1/\sqrt{MK})$ convergence rate by using a quantization function that ensures $q = \Theta(1/m)$.

**Comparison of QSGD and TernGrad** Based on the above results, we now discuss the performance of the quantization functions proposed in the two previous works: QSGD [5] and TernGrad [46]. The quantization function in QSGD has a configurable level $s$. A gradient component $v_i$ is quantized to either $l/s$ or $(l+1)/s$ based on a Bernoulli distribution defined as

$$\begin{cases} \mathbb{P}\{b_i = (l+1)/s\} = \frac{s|v_i|}{\|v\|_2} - l \\ \mathbb{P}\{b_i = l/s\} = 1 - \frac{s|v_i|}{\|v\|_2} + l. \end{cases} \tag{18}$$

The quantized value of $v_i$ is defined as $Q_s(v_i) = b_i \cdot sign(v_i) \cdot \|v\|_2$. Alistarh *et al.* have shown that $Q_s$ has $q \leq \min\{N/s^2, \sqrt{N}/s\}$, while a more accurate bound is $q = \min\{N/(4s^2), \sqrt{N}/s\}$ (see supplemental material B for more explanation). This indicates that $s = \sqrt{N}/2$ is enough to achieve $q = 1$ (thus $\mathcal{O}(1/\sqrt{MK})$ convergence rate) on shared training data. For training with partitioned data (i.e., $\varsigma \neq 0$), we can increase the quantization level of QSGD to $\sqrt{mN}/2$ to achieve $q = 1/m$.

Note that $s$ is not the actual quantization level of $Q_s$, though Alistarh *et al.* called so in their paper [5]. The actual number of different values of $l$ in $Q_s$ is $(s \cdot \|v\|_\infty / \|v\|_2)$. That is, the number of bits used to encode a quantized component is $\lceil \log_2\left(s \cdot \|v\|_\infty / \|v\|_2\right) \rceil$. To see this, we can consider QSGD with $s$ levels in range $[0, \|v\|_2]$ as quantization with $(s \cdot \|v\|_\infty / \|v\|_2)$ levels in range $[0, \|v\|_\infty]$ because both use classification interval of size $\|v\|_2/s$. This also illustrates that the TernGrad proposed by Wen *et al.* [46] is equivalent to QSGD with $s = \|v\|_2 / \|v\|_\infty$ (see supplemental material B and C for more explanation). Therefore, when the gradient components are more evenly distributed (i.e., $s = \|v\|_2 / \|v\|_\infty \to \sqrt{N}$), the fewer quantization levels $Q_s$ needs and the better convergence rate TernGrad can achieve. (Consider the extreme case when the gradient is a vector of same value; It is apparent that quantization level of 1 suffices to encode the gradient.) In general, TernGrad does not achieve $\mathcal{O}(1/\sqrt{MK})$ convergence rate.

# 4   Periodic Quantized Averaging SGD (PQASGD)

---

**Algorithm 2** The procedure on the $i$th node of PQASGD

---

**Require:**  initial point $x_{0,i}$, number of iterations $K$, and learning rate $\gamma$

 1: **for** $j = 0, 1, 2, \ldots, K - 1$ **do**
 2:     Randomly select $m$ training samples indexed by $\xi_{j,i} = [\xi_{j,i,0}, \xi_{j,i,1}, ..., \xi_{j,i,m-1}]$
 3:     Compute a local stochastic gradient based on $\xi_{j,i}$: $\nabla F_i(x_{j,i}; \xi_{j,i})$
 4:     Update the model parameters locally: $x_{j+1,i} = x_{j,i} - \gamma \nabla F_i(x_{j,i}; \xi_{j,i})$
 5:     **if** $((j + 1) \bmod p) = 0$ **then**
 6:         Compute the change of parameters since last synchronization: $G_{j,i} = x_{j+1,i} - x_{j+1-p,i}$
 7:         Quantize the change of parameters: $\Delta_{j,i} = Q(G_{j,i})$
 8:         Average the quantized changes on all nodes: $\Delta_j = \frac{1}{n}\sum_{k=1}^{n} \Delta_{j,k}$
 9:         Update the model parameters: $x_{j+1,i} = x_{j+1-p,i} + \Delta_j$
10:     **end if**
11: **end for**

---

In the previous section, we show that sparse parameter averaging and gradient quantization can achieve $\mathcal{O}(1/\sqrt{MK})$ convergence rate for distributed SGD. However, the compression ratio is limited by using either of the two strategies alone. With sparse parameter averaging, a large $p$ may impair the convergence rate and even lead to divergence. With gradient quantization, even if the gradient components are evenly distributed and the optimal convergence rate can be achieved with 2-bit quantization (one bit for the sign and one bit for the level as in `TernGrad`), the compression ratio is at most $32/2 = 16$ (if no other compression is applied).

We now propose a simple strategy that combines sparsification and quantization to further reduce the communication overhead while preserving the $\mathcal{O}(1/\sqrt{MK})$ convergence rate. The idea is to communicate the quantized changes of model parameters once every $p$ iterations. The procedure is shown in Algorithm 2. All nodes are initialized with the same initial point. In each iteration, each node computes a stochastic gradient based on a local mini-batch and updates its local model parameters. If the iterate number $(j + 1)$ is not a multiple of $p$, the algorithm continues to the next iteration without any communication. If $j + 1$ is a multiple of $p$, each node computes the change of model parameters since last synchronization and quantize the change (Line 6-7). Then, the local quantized changes are averaged among nodes (Line 8) and the average value is updated to the local model parameters (Line 9).

**Convergence Rate**   We have the following convergence rate for Algorithm 2:

**Theorem 3.** *Under the assumptions in §3.1, suppose the quantization function is unbiased with an error bound $q$ and the learning rate $\gamma = \theta\sqrt{M/K}$ where $\theta$ is a constant, Algorithm 2 has the following convergence rate:*

$$\frac{1}{K}\sum_{j=0}^{K-1} \mathbb{E}\left\|\nabla f\left(X_j \frac{\mathbf{1}_n}{n}\right)\right\|_2^2 \leq \frac{2\theta^{-1}\left(f(x_0) - f^*\right)}{\sqrt{MK}} + \frac{2(1 + 2q)L\theta\sigma^2}{\sqrt{MK}} + \frac{12qpmL\theta\varsigma^2}{\sqrt{MK}} \qquad (19)$$

*if the total number of iterations is large enough:*

$$K \geq \max\left(\frac{M\theta^2}{4q^2}\left(\sqrt{(n^2pL)^2 + 12np^2q^2(1 + L^2)} + n^2pL\right)^2, \frac{144ML^2\theta^2q^2p^2}{n^2}, ML^2\theta^2\right) \qquad (20)$$

Theorem 3 implies that `PQASGD` converges at rate $\mathcal{O}((1 + q)/\sqrt{MK})$ if all node share the same training data, and converges at rate $\mathcal{O}(1 + mpq)/\sqrt{MK}$ if the training data are partitioned. Thus, $\mathcal{O}(1/\sqrt{MK})$ convergence rate can be achieved in both cases by using quantization functions that ensure $q = 1$ and $pq = 1/m$ respectively. Compared with stand-alone gradient quantization, `PQASGD` reduces the communication data size by a factor of $p$ on shared training data and a factor of $p/\log(p)$ on partitioned training data.

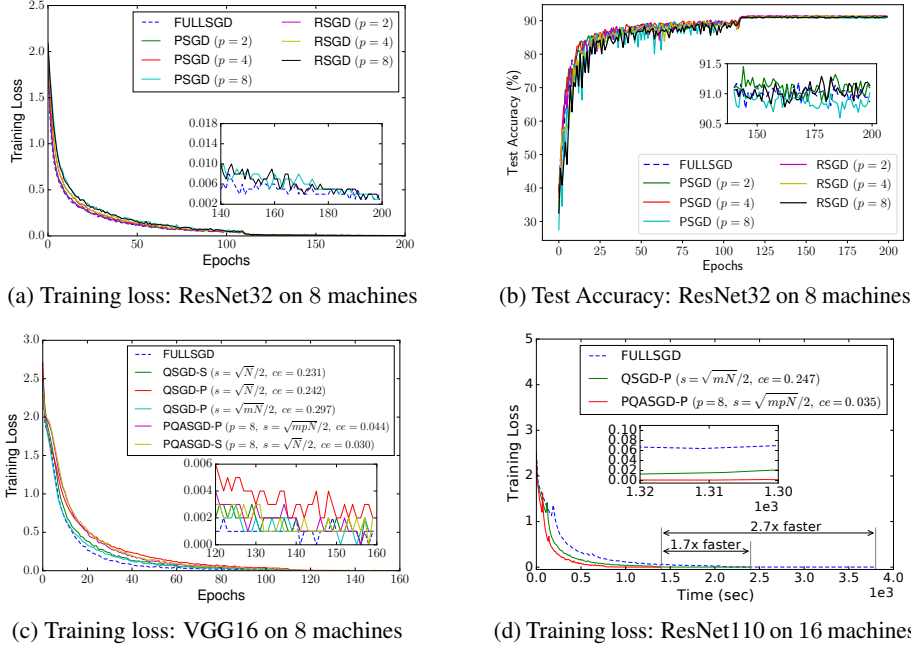

(a) Training loss: ResNet32 on 8 machines

(b) Test Accuracy: ResNet32 on 8 machines

(c) Training loss: VGG16 on 8 machines

(d) Training loss: ResNet110 on 16 machines

Figure 1: Training loss and test accuracy for different image classification models on CIFAR-10.

## 5 Experiments

In this section, we validate our theory with experiments on two machine learning tasks: image classification and speech recognition. For image classification, we train ResNet [19] and VGG [39] with different number of layers on CIFAR-10 [24]. For speech recognition, we train a 5-layer LSTM of $800$ hidden units per layer on AN4 dataset [2].

Our experiments are conducted on a local HPC cluster. Each machine in the cluster has an NVIDIA K80 GPU and is considered as a single node in the training. The machines are connected with 100Gbps InfiniBand and have GPUDirect peer-to-peer communication. We use OpenMPI 3.0.0 as the communication backend, and implement the algorithms in the paper within Pytorch 0.3.1. To make the benefits of communication reduction more noticeable, we direct OpenMPI to use TCP-based communication and use *trickle* to throttle the upload and download bandwidth of the training process on each node to 5Gbps in order to emulate a 10Gbps connection. In practice, network bandwidth can always become a bottleneck when training larger models.

There are three aspects in our evaluation:

*1)* We evaluate two sparse parameter averaging strategies – periodic averaging (PSGD) and rotate averaging (RSGD), on partitioned training data. PSGD averages the model parameters once every $p$ iterations as described in the paper. RSGD divides the parameters into $p$ chunks and averages the $i$th chunk if $j \mod p = i$ where $j$ is the iterate number. It is obvious that RSGD meets the condition in (8). We compare the convergence rate of PSGD and RSGD with full-communication SGD (FULLSGD) to see if they can converge at rate $\mathcal{O}(1/\sqrt{MK})$ as stated in Theorem 1.

*2)* We use QSGD [5] with level $s = \sqrt{N}/2$ on both shared and partitioned training data (QSGD-S and QSGD-P respectively) and compare its convergence rate with FULLSGD. Then, we set the level $s$ to $\sqrt{mN}/2$ for QSGD-P to validate our conclusion in §3.3 that $q = 1/m$ is enough to achieve $\mathcal{O}(1/\sqrt{MK})$ convergence rate for training on partitioned data.

*3)* We compare the convergence rate and performance of our PQASGD with FULLSGD, TernGrad and QSGD. For training on partitioned training data, we set the level $s$ to $\sqrt{mpN}/2$ for PQASGD and $\sqrt{mN}/2$ for QSGD to achieve $q = 1$ for both algorithms.

### 5.1 Results on Image Classification

For all experiments on image classification, we use mini-batch size $m = 32$ on each node. For training on 4 machines (total mini-batch size $M = 128$), we initialize the learning rate $\gamma$ to 0.1 and

decrease it to 0.01 after 110 epochs. The momentum is set to 0.9. For training on more machines, the learning rate scales with the number of nodes. For example, the learning rate for training on 16 machines is initialized to 0.4 and decreased to 0.04 after 110 epochs.

Figure 1a shows the training loss for ResNet32 over epochs on 8 machines with partitioned training data. We can see that PSGD and RSGD with $p$ ranging from 2 to 8 converge almost as fast as FULLSGD. While there is a small gap between the training loss of FULLSGD and PSGD/RSGD with $p = 8$, we can see from the zoomed figure that the gap narrows as the training proceeds. This is observed in all of the models used in our experiments (we have more results in supplemental material D), which validate our claim in Theorem 1.

Figure 1b shows the test accuracy over epochs corresponding to the training process in Figure 1a. We can see that PSGD/RSGD achieve test accuracy comparable to that of FULLSGD, indicating sparse communication does not cause accuracy loss. In fact, we observe that when $p = 2$ and 4, PSGD/RSGD consistently achieve slightly higher accuracy than FULLSGD. As generalization performance of deep neural networks has not been well explained and current theories are mostly based on strong hypotheses [21, 15, 23, 41, 40], we suspect that sparse-communication actually helps the training process escape sharp minimum and avoid overfitting. We will investigate this property of sparse-communication SGD in future work.

Figure 1c shows the training loss for VGG16 over epochs on 8 machines. The number $ce$ in the figure is the *compression efficiency*, which represents the ratio of compressed data size to the original communication data size of FULLSGD. For QSGD, the compression efficiency = $number\_of\_bits\_used/32$. We can see that QSGD-S with $s = \sqrt{N}/2$ matches the convergence rate of FULLSGD with 23.1% communication data size (i.e., with an average of $0.231 \times 32 \approx 7.4$ bits used for each gradient component). In contrast, there is an apparent gap between the training loss of FULLSGD and QSGD-P with $s = \sqrt{N}/2$, which indicates that data partitioning does affect the convergence rate of distributed SGD with quantized gradients. This effect is eliminated by setting $s$ to $\sqrt{mN}/2$. TernGrad achieve 6.3% compression efficiency; however, its training loss after 160 epochs is around 0.02, which is 10 times larger than the other methods. We do not plot the training loss for TernGrad in Figure 1c because it is hard to show its line with other lines in the same scale. From the zoomed figure, we can see that our PQASGD with $p = 8$ matches FULLSGD after 130 epochs on both shared and partitioned training data, while it only incurs 3% and 4.4% communication overhead respectively. Compared with TernGrad, our PQASGD converges much faster while achieving even higher compression ratio. This indicates that instead of simply pursuing more aggressive quantization, combining with sparsification is a more effective approach to reduce the communication overhead for distributed SGD.

Figure 1d shows the training loss for ResNet110 over time on 16 GPUs. We run FULLSGD, QSGD, TernGrad and PQASGD on partitioned training data for 200 epochs. QSGD uses $s = \sqrt{mN}/2$ and PQASGD uses $p = 8$ and $s = \sqrt{mpN}/2$. The mini-batch size and learning rate are the same as described above, except that we set the learning rate to 0.04 in the first 10 epochs for warmup. TernGrad diverges occasionally for training this model, so we do not include its result here. We can see that our PQASGD achieves 2.7x speedup against FULLSGD and 1.7x against QSGD as it requires only 3.5% communication data size compared with FULLSGD. The test error of PQASGD after 200 epochs is 0.0635, which is in consistent with the best accuracy reported in [19]. Thus, our PQASGD does not impair the generalization.

### 5.2 Results on Speech Recognition

The results on speech recognition follow the same pattern as for image classification. Due to space limit, we leave the results in the supplemental material.

## 6 Conclusion

In this work, we studied the convergence rate of distributed SGD with two communication reducing strategies: sparse parameter averaging and gradient quantization. We prove that both strategies can achieve $\mathcal{O}(1/\sqrt{MK})$ convergence rate if configured properly. We also propose a strategy called PQASGD that combines sparsification and quantization while preserving the $\mathcal{O}(1/\sqrt{MK})$ convergence rate. The experiments validate our theoretical results and show that our PQASGD matches the convergence rate of full-communication SGD with only 3%-5% communication data size.

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
