[Supplementary Material]

# Supplemental Materials

## A    Proofs to theorems in the paper

We start by listing the basic inequalities that are frequently used in our proof.

**Inequality 1.**

$$\left\| \sum_{i=1}^{n} (x_i - y_i) \right\|_2^2 = \sum_{i=1}^{n} \sum_{j=1}^{n} \langle x_i - y_i, x_j - y_j \rangle$$

$$= \sum_{i=1}^{n} \sum_{j=1}^{n} \frac{1}{2} \left( \|x_i - y_i\|_2^2 + \|x_j - y_j\|_2^2 - \|(x_i - y_i) - (x_j - y_j)\|_2^2 \right)$$

$$\leq n \sum_{i=1}^{n} \|x_i - y_i\|_2^2 \tag{21}$$

If

$$\mathbb{E}[x_i - y_i] = 0, \forall i,$$

then

$$\mathbb{E} \left\| \sum_{i=1}^{n} (x_i - y_i) \right\|_2^2 = \sum_{i=1}^{n} \mathbb{E}\|x_i - y_i\|_2^2 + \sum_{i \neq j}^{n} \mathbb{E} \langle x_i - y_i, x_j - y_j \rangle = \sum_{i=1}^{n} \mathbb{E}\|x_i - y_i\|_2^2 \tag{22}$$

**Inequality 2.** From the Lipschitzan gradient assumption in (3) and Inequality in (21), we can obtain:

$$\left\|\nabla f(x) - \nabla f(y)\right\|_2^2 = \left\| \frac{1}{n} \sum_{i=1}^{n} \left( \nabla f_i(x) - \nabla f_i(y) \right) \right\|_2^2 \leq \frac{1}{n} \sum_{i=1}^{n} \left\| \nabla f_i(x) - \nabla f_i(y) \right\|_2^2 \leq L^2 \|x - y\|_2^2 \tag{23}$$

Thus,

$$\left\|\nabla f(x) - \nabla f(y)\right\|_2 \leq L\|x - y\|_2 \tag{24}$$

which is equivalent to

$$f(y) \leq f(x) + \nabla f(x)^T (y - x) + \frac{L}{2} \|y - x\|_2^2. \tag{25}$$

To see this, let us consider a function $g(x) = \frac{L}{2} x^T x - f(x)$. Because $\nabla g(x) = Lx - \nabla f(x)$, we have $(x - y) \left( \nabla g(x) - \nabla g(y) \right) = L\|x - y\|_2^2 - \left( \nabla f(x) - \nabla f(y) \right)^T (x - y) \geq 0$, which implies $g(x)$ is convex. Note that $f(x)$ can be non-convex. It follows that $g(y) \geq g(x) + \nabla g(x)^T (y - x)$, i.e., $\frac{L}{2} y^T y - f(y) \geq \frac{L}{2} x^T x - f(x) + \left( Lx - \nabla f(x) \right)^T (y - x)$, which is equivalent to (25).

***Proof to Theorem 1.*** The computation of Algorithm 1 can be expressed as

$$X_{j+1} = \mathcal{P}_j(X_j - \gamma \partial F(X_j; \xi_j))W_n + (\mathcal{I} - \mathcal{P}_j)(X_j - \gamma \partial F(X_j; \xi_j)) \tag{26}$$

Here, $X_j = [x_{j,1}, x_{j,2}, ..., x_{j,n}] \in \mathbb{R}^{N \times n}$ are the model parameters on $n$ nodes, $\mathcal{P}_j$ is the projection matrix representing selected components as defined in the algorithm, $W_n$ is a $n$ by $n$ matrix with all elements being $1/n$, and $\partial F(X_j; \xi_j) = [\nabla F_1(x_{j,1}; \xi_{j,1}), \nabla F_2(x_{j,2}; \xi_{j,2}), ..., \nabla F_n(x_{j,n}; \xi_{j,n})]$ are the stochastic gradients. Our goal is to bound the average squared norm of gradients over $K$ iterations, i.e.,

$$\frac{1}{K} \left( \sum_{j=0}^{K-1} \left\| \nabla f \left( \frac{X_j \mathbf{1_n}}{n} \right) \right\|_2^2 \right).$$

From (26) and because $W_n \frac{\mathbf{1}_n}{n} = \frac{\mathbf{1}_n}{n}$, we have:

$$\mathbb{E}f \left( X_{j+1} \frac{\mathbf{1}_n}{n} \right) = \mathbb{E}f \left( \mathcal{P}_j(X_j - \gamma \partial F(X_j; \xi_j)) \frac{\mathbf{1}_n}{n} + (\mathcal{I} - \mathcal{P}_j)(X_j - \gamma \partial F(X_j; \xi_j)) \frac{\mathbf{1}_n}{n} \right)$$

$$= \mathbb{E}f \left( \left( X_j - \gamma \partial F(X_j; \xi_j) \right) \frac{\mathbf{1}_n}{n} \right) \tag{27}$$

Then, we apply Lipschitzan gradient assumption in (25) to (27) and obtain:

$$\mathbb{E}f\left(X_{j+1}\frac{\mathbf{1}_n}{n}\right) \leq \mathbb{E}f\left(X_j\frac{\mathbf{1}_n}{n}\right) - \underbrace{\mathbb{E}\left\langle \nabla f\left(X_j\frac{\mathbf{1}_n}{n}\right), \gamma \partial F\left(X_j;\xi_j\right)\frac{\mathbf{1}_n}{n}\right\rangle}_{=:T_1} + \frac{L}{2}\underbrace{\mathbb{E}\left\|\gamma \partial F(X_j,\xi_j)\frac{\mathbf{1}_n}{n}\right\|_2^2}_{=:T_2}$$

(28)

For $T_1$, we have:

$$T_1 := \mathbb{E}\left\langle \nabla f\left(X_j\frac{\mathbf{1}_n}{n}\right), \gamma \partial F\left(X_j;\xi_j\right)\frac{\mathbf{1}_n}{n}\right\rangle$$

$$\overset{\because \text{Assumption (4)}}{=} \gamma \mathbb{E}\left\langle \nabla f\left(X_j\frac{\mathbf{1}_n}{n}\right), \partial f\left(X_j\right)\frac{\mathbf{1}_n}{n}\right\rangle$$

$$= \frac{\gamma}{2}\mathbb{E}\left\|\nabla f\left(X_j\frac{\mathbf{1}_n}{n}\right)\right\|_2^2 + \frac{\gamma}{2}\mathbb{E}\left\|\partial f\left(X_j\right)\frac{\mathbf{1}_n}{n}\right\|_2^2 - \frac{\gamma}{2}\underbrace{\mathbb{E}\left\|\nabla f\left(X_j\frac{\mathbf{1}_n}{n}\right) - \partial f(X_j)\frac{\mathbf{1}_n}{n}\right\|_2^2}_{=:T_3}$$

(29)

$$T_3 := \mathbb{E}\left\|\nabla f\left(X_j\frac{\mathbf{1}_n}{n}\right) - \partial f(X_j)\frac{\mathbf{1}_n}{n}\right\|_2^2 = \mathbb{E}\left\|\frac{1}{n}\sum_{i=1}^n \nabla f_i\left(X_j\frac{\mathbf{1}_n}{n}\right) - \frac{1}{n}\sum_{i=1}^n \nabla f_i\left(X_j\mathbf{e}_i\right)\right\|_2^2$$

$$\overset{\text{According to (21)}}{\leq} \frac{1}{n}\sum_{i=1}^n \mathbb{E}\left\|\nabla f_i\left(X_j\frac{\mathbf{1}_n}{n}\right) - \nabla f_i\left(X_j\mathbf{e}_i\right)\right\|_2^2$$

$$\overset{\text{Assumption (3)}}{\leq} \frac{L^2}{n}\underbrace{\sum_{i=1}^n \mathbb{E}\left\|X_j\frac{\mathbf{1}_n}{n} - X_j\mathbf{e}_i\right\|_2^2}_{=:T_{4j}}$$

(30)

If we plug $T_3$ and $T_1$ back to (28), we can get:

$$\mathbb{E}f\left(X_{j+1}\frac{\mathbf{1}_n}{n}\right) \leq \mathbb{E}f\left(X_j\frac{\mathbf{1}_n}{n}\right) - \frac{\gamma}{2}\mathbb{E}\left\|\nabla f\left(X_j\frac{\mathbf{1}_n}{n}\right)\right\|_2^2 - \frac{\gamma}{2}\mathbb{E}\left\|\partial f\left(X_j\right)\frac{\mathbf{1}_n}{n}\right\|_2^2$$

$$+ \frac{L^2\gamma}{2}\boxed{\frac{1}{n}\sum_{i=1}^n \mathbb{E}\left\|X_j\frac{\mathbf{1}_n}{n} - X_j\mathbf{e}_i\right\|_2^2} + \frac{L\gamma^2}{2}\mathbb{E}\left\|\partial F(X_j,\xi_j)\frac{\mathbf{1}_n}{n}\right\|_2^2$$

(31)

Note that the term in the box is the variance of model parameters among nodes. We can see that the smaller the variance of model parameters among nodes, the smaller the bound of the squared norm of gradient at iteration $j$. Intuitively, a smaller variance of model parameters among nodes indicates that the trajectories of training on different nodes are not far apart, and thus, leads to better convergence of the algorithm. Therefore, the key to our proof is to bound the variance of model parameters (i.e., $T_{4j}/n$).

From (26), we can derive

$$X_j\left(\frac{\mathbf{1}_n}{n} - \mathbf{e}_i\right) = (\mathcal{I} - \mathcal{P}_{j-1})(X_{j-1} - \gamma \partial F(X_{j-1};\xi_{j-1}))\left(\frac{\mathbf{1}_n}{n} - \mathbf{e}_i\right)$$

$$= (\mathcal{I} - \mathcal{P}_{j-1})\left((\mathcal{I} - \mathcal{P}_{j-2})(X_{j-2} - \gamma \partial F(X_{j-2};\xi_{j-2}))\right)\left(\frac{\mathbf{1}_n}{n} - \mathbf{e}_i\right)$$

$$- \gamma(\mathcal{I} - \mathcal{P}_{j-1})\partial F(X_{j-1};\xi_{j-1})\left(\frac{\mathbf{1}_n}{n} - \mathbf{e}_i\right)$$

$$= \left(\prod_{t=0}^{j-1}(\mathcal{I} - \mathcal{P}_{j-1-t})X_0 - \gamma \sum_{k=0}^{j-1}\prod_{t=0}^k (\mathcal{I} - \mathcal{P}_{j-1-t})\partial F(X_{j-1-k};\xi_{j-1-k})\right)\left(\frac{\mathbf{1}_n}{n} - \mathbf{e}_i\right)$$

$$\overset{\because x_{0,i}=x_0,\forall i}{=} \gamma \sum_{k=0}^{j-1}\prod_{t=0}^k (\mathcal{I} - \mathcal{P}_{j-1-t})\partial F(X_{j-1-k};\xi_{j-1-k})\left(\frac{\mathbf{1}_n}{n} - \mathbf{e}_i\right)$$

(32)

Therefore, we have

$$
T_{4j} := \sum_{i=1}^{n} \mathbb{E} \left\| X_j \frac{\mathbf{1}_n}{n} - X_j \mathbf{e}_i \right\|_2^2
$$

$$
= \gamma^2 \sum_{i=1}^{n} \mathbb{E} \left\| \sum_{k=0}^{j-1} \prod_{t=0}^{k} (\mathcal{I} - \mathcal{P}_{j-1-t}) \partial F(X_{j-1-k}; \xi_{j-1-k}) \left( \frac{\mathbf{1}_n}{n} - \mathbf{e}_i \right) \right\|_2^2
$$

$$
= \gamma^2 \sum_{i=1}^{n} \mathbb{E} \left\| \sum_{k=0}^{j-1} \prod_{t=0}^{k} (\mathcal{I} - \mathcal{P}_{j-1-t}) \left( \partial F(X_{j-1-k}; \xi_{j-1-k}) - \partial f(X_{j-1-k}) + \partial f(X_{j-1-k}) \right) \left( \frac{\mathbf{1}_n}{n} - \mathbf{e}_i \right) \right\|_2^2
$$

$$
= \gamma^2 \sum_{i=1}^{n} \mathbb{E} \left\| \sum_{k=0}^{j-1} \prod_{t=0}^{k} (\mathcal{I} - \mathcal{P}_{j-1-t}) \left( \partial F(X_{j-1-k}; \xi_{j-1-k}) - \partial f(X_{j-1-k}) \right) \left( \frac{\mathbf{1}_n}{n} - \mathbf{e}_i \right) \right\|_2^2
$$

$$
+ \gamma^2 \sum_{i=1}^{n} \mathbb{E} \left\| \sum_{k=0}^{j-1} \prod_{t=0}^{k} (\mathcal{I} - \mathcal{P}_{j-1-t}) \partial f(X_{j-1-k}) \left( \frac{\mathbf{1}_n}{n} - \mathbf{e}_i \right) \right\|_2^2
$$

$$
\overset{\because \text{Condition in (8)}}{\leq} \gamma^2 \sum_{i=1}^{n} \mathbb{E} \left\| \sum_{k=0}^{p-1} \prod_{t=0}^{k} (\mathcal{I} - \mathcal{P}_{j-1-t}) \left( \partial F(X_{j-1-k}; \xi_{j-1-k}) - \partial f(X_{j-1-k}) \right) \left( \frac{\mathbf{1}_n}{n} - \mathbf{e}_i \right) \right\|_2^2
$$

$$
+ \gamma^2 \sum_{i=1}^{n} \mathbb{E} \left\| \sum_{k=0}^{p-1} \prod_{t=0}^{k} (\mathcal{I} - \mathcal{P}_{j-1-t}) \partial f(X_{j-1-k}) \left( \frac{\mathbf{1}_n}{n} - \mathbf{e}_i \right) \right\|_2^2
$$

$$
\leq \gamma^2 \sum_{i=1}^{n} \mathbb{E} \left\| \sum_{k=0}^{p-1} \left( \partial F(X_{j-1-k}; \xi_{j-1-k}) - \partial f(X_{j-1-k}) \right) \left( \frac{\mathbf{1}_n}{n} - \mathbf{e}_i \right) \right\|_2^2
$$

$$
+ \gamma^2 \sum_{i=1}^{n} \mathbb{E} \left\| \sum_{k=0}^{p-1} \partial f(X_{j-1-k}) \left( \frac{\mathbf{1}_n}{n} - \mathbf{e}_i \right) \right\|_2^2
$$

$$
\overset{\text{According to (21) and (22)}}{\leq} \gamma^2 \sum_{i=1}^{n} \sum_{k=0}^{p-1} \underbrace{\mathbb{E} \left\| \left( \partial F(X_{j-1-k}; \xi_{j-1-k}) - \partial f(X_{j-1-k}) \right) \left( \frac{\mathbf{1}_n}{n} - \mathbf{e}_i \right) \right\|_2^2}_{=:T_5}
$$

$$
+ \gamma^2 p \sum_{i=1}^{n} \sum_{k=0}^{p-1} \underbrace{\mathbb{E} \left\| \partial f(X_{j-1-k}) \left( \frac{\mathbf{1}_n}{n} - \mathbf{e}_i \right) \right\|_2^2}_{=:T_6}
$$

(33)

$$
T_5 := \mathbb{E} \left\| \left( \partial F(X_{j-1-k}; \xi_{j-1-k}) - \partial f(X_{j-1-k}) \right) \left( \frac{\mathbf{1}_n}{n} - \mathbf{e}_i \right) \right\|_2^2
$$

$$
\leq \mathbb{E} \left\| \partial F(X_{j-1-k}; \xi_{j-1-k}) - \partial f(X_{j-1-k}) \right\|_F^2
$$

$$
= \sum_{i=1}^{n} \mathbb{E} \left\| \nabla F_i(x_{j-1-k,i}; \xi_{j-1-k,i}) - \nabla f_i(x_{j-1-k,i}) \right\|_2^2
$$

$$
= \sum_{i=1}^{n} \mathbb{E} \left\| \frac{1}{m} \sum_{s=1}^{m} \nabla F_i(x_{j-1-k,i}; \xi_{j-1-k,i,s}) - \nabla f_i(x_{j-1-k,i}) \right\|_2^2
$$

$$
\overset{\text{According to (22)}}{=} \frac{1}{m^2} \sum_{i=1}^{n} \sum_{s=1}^{m} \mathbb{E} \left\| \nabla F_i(x_{j-1-k,i}; \xi_{j-1-k,i,s}) - \nabla f_i(x_{j-1-k,i}) \right\|_2^2
$$

$$
\leq \frac{n \sigma^2}{m}
$$

(34)

$$T_6 := \mathbb{E}\left\|\partial f(X_{j-1-k})\left(\frac{\mathbf{1}_n}{n} - \mathbf{e}_i\right)\right\|_2^2$$

$$\leq \mathbb{E}\left\|\partial f(X_{j-1-k})\right\|_F^2 = \sum_{i=1}^n \mathbb{E}\left\|\nabla f_i(x_{j-1-k,i})\right\|_2^2$$

$$= \sum_{i=1}^n \mathbb{E}\left\|\nabla f_i(x_{j-1-k,i}) - \nabla f(x_{j-1-k,i}) + \nabla f(x_{j-1-k,i}) - \nabla f\left(X_{j-1-k}\frac{\mathbf{1}_n}{n}\right) + \nabla f\left(X_{j-1-k}\frac{\mathbf{1}_n}{n}\right)\right\|_2^2$$

$$\overset{\text{According to (21)}}{\leq} 3\sum_{i=1}^n \mathbb{E}\left\|\nabla f_i(x_{j-1-k,i}) - \nabla f(x_{j-1-k,i})\right\|_2^2 + 3\sum_{i=1}^n \mathbb{E}\left\|\nabla f(x_{j-1-k,i}) - \nabla f\left(X_{j-1-k}\frac{\mathbf{1}_n}{n}\right)\right\|_2^2$$

$$+ 3\sum_{i=1}^n \mathbb{E}\left\|\nabla f\left(X_{j-1-k}\frac{\mathbf{1}_n}{n}\right)\right\|_2^2$$

$$\overset{\because \text{Assumptions (6) and (3)}}{\leq} 3n\varsigma^2 + 3L^2 \underbrace{\sum_{i=1}^n \mathbb{E}\left\|x_{j-1-k,i} - X_{j-1-k}\frac{\mathbf{1}_n}{n}\right\|_2^2}_{=:T_{4j-1-k}} + 3n\mathbb{E}\left\|\nabla f\left(X_{j-1-k}\frac{\mathbf{1}_n}{n}\right)\right\|_2^2 \tag{35}$$

From (33), (34) and (35), we have the bound for $T_{4j}$:

$$T_{4j} \leq \frac{p\gamma^2 n^2 \sigma^2}{m} + 3p^2\gamma^2 n^2\varsigma^2 + 3pnL^2\gamma^2 \sum_{k=0}^{p-1} T_{4j-1-k} + 3pn^2\gamma^2 \sum_{k=0}^{p-1} \mathbb{E}\left\|\nabla f\left(X_{j-1-k}\frac{\mathbf{1}_n}{n}\right)\right\|_2^2 \tag{36}$$

It follows that

$$\sum_{j=0}^{K-1} T_{4j} \leq \frac{p\gamma^2 n^2\sigma^2 K}{m} + 3p^2\gamma^2 n^2\varsigma^2 K + 3np^2 L^2\gamma^2 \sum_{j=0}^{K-1} T_{4j} + 3p^2 n^2\gamma^2 \sum_{j=0}^{K-1} \mathbb{E}\left\|\nabla f\left(X_j\frac{\mathbf{1}_n}{n}\right)\right\|_2^2 \tag{37}$$

Suppose

$$1 > 3np^2 L^2\gamma^2 \tag{38}$$

we have

$$\sum_{j=0}^{K-1} T_{4j} \leq \frac{pn^2\gamma^2 \left(K\left(\frac{\sigma^2}{m} + 3p\varsigma^2\right) + 3p \sum_{j=0}^{K-1} \mathbb{E}\left\|\nabla f\left(X_j\frac{\mathbf{1}_n}{n}\right)\right\|_2^2\right)}{1 - 3np^2 L^2\gamma^2} \tag{39}$$

For $T_2$, we have

$$T_2 := \mathbb{E}\left\|\gamma \partial F(X_j, \xi_j)\frac{\mathbf{1}_n}{n}\right\|_2^2$$

$$= \gamma^2 \mathbb{E}\left\|\partial F(X_j, \xi_j)\frac{\mathbf{1}_n}{n} - \partial f(X_j)\frac{\mathbf{1}_n}{n} + \partial f(X_j)\frac{\mathbf{1}_n}{n}\right\|_2^2$$

$$= \gamma^2 \mathbb{E}\left\|\partial F(X_j, \xi_j)\frac{\mathbf{1}_n}{n} - \partial f(X_j)\frac{\mathbf{1}_n}{n}\right\|_2^2 + \gamma^2 \mathbb{E}\left\|\partial f(X_j)\frac{\mathbf{1}_n}{n}\right\|_2^2$$

$$= \gamma^2 \mathbb{E}\left\|\frac{1}{mn}\sum_{i=1}^n \sum_{s=1}^m \nabla F_i(x_{j,i}, \xi_{j,i,s}) - \nabla f_i(x_{j,i})\right\|_2^2 + \gamma^2 \mathbb{E}\left\|\partial f(X_j)\frac{\mathbf{1}_n}{n}\right\|_2^2$$

$$\overset{\text{According to (22)}}{\leq} \frac{\gamma^2\sigma^2}{mn} + \gamma^2 \mathbb{E}\left\|\partial f(X_j)\frac{\mathbf{1}_n}{n}\right\|_2^2 \tag{40}$$

From (27), (29), (30) and (40), we can obtain

$$\mathbb{E}f\left(X_{j+1}\frac{\mathbf{1}_n}{n}\right) \leq \mathbb{E}f\left(X_j\frac{\mathbf{1}_n}{n}\right) - \frac{\gamma}{2}\mathbb{E}\left\|\nabla f\left(X_j\frac{\mathbf{1}_n}{n}\right)\right\|_2^2 - \frac{\gamma}{2}\mathbb{E}\left\|\partial f(X_j)\frac{\mathbf{1}_n}{n}\right\|_2^2 + \frac{L^2\gamma}{2n}T_{4j}$$

$$+ \frac{L\gamma^2\sigma^2}{2mn} + \frac{L\gamma^2}{2}\mathbb{E}\left\|\partial f(X_j)\frac{\mathbf{1}_n}{n}\right\|_2^2 \tag{41}$$

Suppose

$$\frac{\gamma}{2} \geq \frac{L\gamma^2}{2} \tag{42}$$

we have

$$\mathbb{E}f\left(X_{j+1}\frac{\mathbf{1}_n}{n}\right) \leq \mathbb{E}f\left(X_j\frac{\mathbf{1}_n}{n}\right) - \frac{\gamma}{2}\mathbb{E}\left\|\nabla f\left(X_j\frac{\mathbf{1}_n}{n}\right)\right\|_2^2 + \frac{L^2\gamma}{2n}T_{4j} + \frac{L\gamma^2\sigma^2}{2mn} \tag{43}$$

Summing up from $\mathbb{E}f(x_1)$ to $\mathbb{E}f(x_K)$, we have:

$$\mathbb{E}f(x_K) \leq \mathbb{E}f(x_0) - \frac{\gamma}{2}\sum_{j=0}^{K-1}\mathbb{E}\left\|\nabla f\left(X_j\frac{\mathbf{1}_n}{n}\right)\right\|_2^2 + \frac{L^2\gamma}{2n}\sum_{j=0}^{K-1}T_{4j} + \frac{L\gamma^2\sigma^2 K}{2mn} \tag{44}$$

Plugging (39) into (45), we can obtain

$$\mathbb{E}f(x_K) \leq \mathbb{E}f(x_0) - \frac{\gamma}{2}\sum_{j=0}^{K-1}\mathbb{E}\left\|\nabla f\left(X_j\frac{\mathbf{1}_n}{n}\right)\right\|_2^2 + \frac{L^2\gamma}{2n} \cdot \frac{pn^2\gamma^2 K\left(\frac{\sigma^2}{m} + 3p\varsigma^2\right)}{1 - 3np^2L^2\gamma^2}$$
$$+ \frac{L^2\gamma}{2n} \cdot \frac{3p^2n^2\gamma^2}{1 - 3np^2L^2\gamma^2}\sum_{j=0}^{K-1}\mathbb{E}\left\|\nabla f\left(X_j\frac{\mathbf{1}_n}{n}\right)\right\|_2^2 + \frac{L\gamma^2\sigma^2 K}{2mn} \tag{45}$$

Suppose

$$\frac{\gamma}{2} > \frac{L^2\gamma}{2n} \cdot \frac{3p^2n^2\gamma^2}{1 - 3np^2L^2\gamma^2} \tag{46}$$

i.e.,

$$1 > 6np^2L^2\gamma^2, \tag{47}$$

then it follows from (45) that

$$\frac{1}{K}\left(\sum_{j=0}^{K-1}\left\|\nabla f\left(\frac{X_j\mathbf{1}_n}{n}\right)\right\|_2^2\right) \leq \frac{2\left(f(x_0) - f^*\right)D_1}{\gamma K} + \frac{L\gamma\sigma^2 D_1}{M} + \frac{pn^2L^2\gamma^2\sigma^2 D_2}{M} + 3np^2L^2\gamma^2\varsigma^2 D_2, \tag{48}$$

where

$$D_1 = \left(\frac{1 - 3np^2L^2\gamma^2}{1 - 6np^2L^2\gamma^2}\right), \quad D_2 = \left(\frac{1}{1 - 6np^2L^2\gamma^2}\right). \tag{49}$$

The above bound holds when (38), (42) and (47) are satisfied, which is Theorem 1. $\quad\square$

If we set $\gamma = \theta\sqrt{M/K}$ and let $D_1 \leq 2$ and $D_2 \leq 2$, i.e.,

$$1 \geq 12np^2L^2\gamma^2 \iff K \geq 12nMp^2L^2\theta^2,$$

we can obtain Corollary 1 from Theorem 1.

***Proof to Theorem 2.*** The computation of distributed SGD with gradient quantization can be expressed as

$$x_{j+1} = x_j - \gamma Q\left(\partial F(X_j; \xi_j)\right)\frac{\mathbf{1}_n}{n} \tag{50}$$

Here, $X_j = [x_{j,1}, x_{j,2}, ..., x_{j,n}] \in \mathbb{R}^{N \times n}$ are the model parameters on $n$ nodes, $Q$ is any unbiased stochastic quantization function, and $\partial F(X_j; \xi_j) = [\nabla F_1(x_{j,1}; \xi_{j,1}), \nabla F_2(x_{j,2}; \xi_{j,2}), ..., \nabla F_n(x_{j,n}; \xi_{j,n})]$ are the stochastic gradients. Note that in this algorithm $x_{j,i} = x_j, \forall j, \forall i$.

Consider $\mathbb{E}f(x_{+1})$:

$$\mathbb{E}f(x_{j+1}) = \mathbb{E}f\left(x_j - \gamma Q\left(\partial F(X_j; \xi_j)\right)\frac{\mathbf{1}_n}{n}\right)$$

$$\overset{\text{According to (25)}}{\leq} \mathbb{E}f(x_j) - \underbrace{\mathbb{E}\left\langle \nabla f(x_j), \gamma Q\left(\partial F(X_j; \xi_j)\right)\frac{\mathbf{1}_n}{n}\right\rangle}_{=:T_1} + \frac{L}{2}\underbrace{\mathbb{E}\left\|\gamma Q\left(\partial F(X_j; \xi_j)\right)\frac{\mathbf{1}_n}{n}\right\|_2^2}_{=:T_2} \tag{51}$$

For $T_1$ and $T_2$, we have:

$$\begin{aligned}
T_1 :&= \mathbb{E}\left\langle \nabla f(x_j), \gamma Q\left(\partial F(X_j; \xi_j)\right)\frac{\mathbf{1}_n}{n}\right\rangle\\
&\overset{\text{Law of total expectation}}{=} \gamma\mathbb{E}\left[\mathbb{E}_{Q\sim\mathcal{H}}\left\langle \nabla f(x_j), Q\left(\partial F(X_j; \xi_j)\right)\frac{\mathbf{1}_n}{n}\right\rangle\right]\\
&\overset{Q \text{ is unbiased}}{=} \gamma\mathbb{E}\left\langle \nabla f(x_j), \partial F(X_j; \xi_j)\frac{\mathbf{1}_n}{n}\right\rangle\\
&= \gamma\mathbb{E}\left[\mathbb{E}_{\xi_j\sim\mathcal{D}}\left\langle \nabla f(x_j), \partial F(X_j; \xi_j)\frac{\mathbf{1}_n}{n}\right\rangle\right]\\
&\overset{\text{Assumption (4)}}{=} \gamma\mathbb{E}\left\langle \nabla f(x_j), \partial f(X_j)\frac{\mathbf{1}_n}{n}\right\rangle\\
&\overset{x_{j,i}=x_j, \forall i}{=} \gamma\mathbb{E}\left\langle \nabla f(x_j), \frac{1}{n}\sum_{i=1}^n \nabla f_i(x_j)\right\rangle \overset{\text{According to (2)}}{=} \gamma\mathbb{E}\left\|\nabla f(x_j)\right\|_2^2
\end{aligned} \tag{52}$$

$$\begin{aligned}
T_2 :&= \mathbb{E}\left\|\gamma Q\left(\partial F(X_j; \xi_j)\right)\frac{\mathbf{1}_n}{n}\right\|_2^2\\
&= \gamma^2\mathbb{E}\left\|Q\left(\partial F(X_j; \xi_j)\right)\frac{\mathbf{1}_n}{n}\right\|_2^2\\
&= \gamma^2\mathbb{E}\left\|Q\left(\partial F(X_j; \xi_j)\right)\frac{\mathbf{1}_n}{n} - \partial F(X_j; \xi_j)\frac{\mathbf{1}_n}{n} + \partial F(X_j; \xi_j)\frac{\mathbf{1}_n}{n}\right\|_2^2\\
&= \gamma^2\mathbb{E}\left\|Q\left(\partial F(X_j; \xi_j)\right)\frac{\mathbf{1}_n}{n} - \partial F(X_j; \xi_j)\frac{\mathbf{1}_n}{n}\right\|_2^2 + \gamma^2\mathbb{E}\left\|\partial F(X_j; \xi_j)\frac{\mathbf{1}_n}{n}\right\|_2^2\\
&\quad + \gamma^2\mathbb{E}\left\langle Q\left(\partial F(X_j; \xi_j)\right)\frac{\mathbf{1}_n}{n} - \partial F(X_j; \xi_j)\frac{\mathbf{1}_n}{n}, \partial F(X_j; \xi_j)\frac{\mathbf{1}_n}{n}\right\rangle\\
&\overset{\because Q \text{ is unbiased}}{=} \gamma^2\underbrace{\mathbb{E}\left\|Q\left(\partial F(X_j; \xi_j)\right)\frac{\mathbf{1}_n}{n} - \partial F(X_j; \xi_j)\frac{\mathbf{1}_n}{n}\right\|_2^2}_{=:T_3} + \gamma^2\underbrace{\mathbb{E}\left\|\partial F(X_j; \xi_j)\frac{\mathbf{1}_n}{n}\right\|_2^2}_{=:T_4}
\end{aligned} \tag{53}$$

$$\begin{aligned}
T_3 =& \mathbb{E}\left\|\frac{1}{n}\sum_{i=1}^n\left(Q\left(\nabla F_i(x_j; \xi_{j,i})\right) - \nabla F_i(x_j; \xi_{j,i})\right)\right\|_2^2\\
&\overset{\text{According to (22)}}{=} \frac{1}{n^2}\sum_{i=1}^n\mathbb{E}\left\|Q\left(\nabla F_i(x_j; \xi_{j,i})\right) - \nabla F_i(x_j; \xi_{j,i})\right\|_2^2\\
&\overset{\text{According to (13)}}{\leq} \frac{q}{n^2}\sum_{i=1}^n\mathbb{E}\left\|\nabla F_i(x_j; \xi_{j,i})\right\|_2^2 = \frac{q}{n^2}\sum_{i=1}^n\mathbb{E}\left\|\nabla F_i(x_j; \xi_{j,i}) - \nabla f_i(x_j) + \nabla f_i(x_j)\right\|_2^2\\
&= \frac{q}{n^2}\sum_{i=1}^n\mathbb{E}\left\|\nabla F_i(x_j; \xi_{j,i}) - \nabla f_i(x_j)\right\|_2^2 + \frac{q}{n^2}\sum_{i=1}^n\mathbb{E}\left\|\nabla f_i(x_j)\right\|_2^2\\
&= \frac{q}{n^2}\sum_{i=1}^n\mathbb{E}\left\|\frac{1}{m}\sum_{s=1}^m\nabla F_i(x_j; \xi_{j,i,s}) - \nabla f_i(x_j)\right\|_2^2 + \frac{q}{n^2}\sum_{i=1}^n\mathbb{E}\left\|\nabla f_i(x_j)\right\|_2^2\\
&\overset{\text{Assumption in (5)}}{\leq} \frac{q\sigma^2}{mn} + \frac{q}{n^2}\sum_{i=1}^n\mathbb{E}\left\|\nabla f_i(x_j) - \nabla f(x_j) + \nabla f(x_j)\right\|_2^2\\
&\overset{\because \nabla f(x)=\frac{1}{n}\sum_i\nabla f_i(x)}{=} \frac{q\sigma^2}{mn} + \frac{q}{n^2}\sum_{i=1}^n\mathbb{E}\left\|\nabla f_i(x_j) - \nabla f(x_j)\right\|_2^2 + \frac{q}{n^2}\sum_{i=1}^n\mathbb{E}\left\|\nabla f(x_j)\right\|_2^2
\end{aligned}$$

$$+ \frac{q}{n^2} \sum_{i=1}^{n} \mathbb{E} \left\langle \nabla f_i(x_j) - \nabla f(x_j), \nabla f(x_j) \right\rangle$$

$$= \frac{q\sigma^2}{mn} + \frac{q}{n^2} \sum_{i=1}^{n} \mathbb{E} \left\| \nabla f_i(x_j) - \nabla f(x_j) \right\|_2^2 + \frac{q}{n^2} \sum_{i=1}^{n} \mathbb{E} \left\| \nabla f(x_j) \right\|_2^2$$

$$\leq \frac{q\sigma^2}{mn} + \frac{q\varsigma^2}{n} + \frac{q}{n} \mathbb{E} \left\| \nabla f(x_j) \right\|_2^2 \tag{54}$$

$$
\begin{aligned}
T_4 &= \mathbb{E} \left\| \partial F(X_j; \xi_j) \frac{\mathbf{1}_n}{n} - \partial f(X_j) \frac{\mathbf{1}_n}{n} + \partial f(X_j) \frac{\mathbf{1}_n}{n} \right\|_2^2 \\
&= \mathbb{E} \left\| \partial F(X_j; \xi_j) \frac{\mathbf{1}_n}{n} - \partial f(X_j) \frac{\mathbf{1}_n}{n} \right\|_2^2 + \mathbb{E} \left\| \partial f(X_j) \frac{\mathbf{1}_n}{n} \right\|_2^2 \\
&= \mathbb{E} \left\| \frac{1}{n} \sum_{i=1}^{n} \left( \nabla F_i(x_j; \xi_{j,i}) - \nabla f_i(x_j) \right) \right\|_2^2 + \mathbb{E} \left\| \nabla f(x_j) \right\|_2^2 \\
&\leq \frac{\sigma^2}{mn} + \mathbb{E} \left\| \nabla f(x_j) \right\|_2^2
\end{aligned} \tag{55}
$$

Plugging $T_1, T_2$ into (51), we have:

$$
\begin{aligned}
\mathbb{E} f(x_{j+1}) &\leq \mathbb{E} f(x_j) - \gamma \mathbb{E} \left\| \nabla f(x_j) \right\|_2^2 + \frac{(1+q)L\gamma^2 \sigma^2}{2mn} + \frac{qL\gamma^2 \varsigma^2}{2n} + \frac{(q+n)L\gamma^2}{2n} \mathbb{E} \left\| \nabla f(x_j) \right\|_2^2 \\
&= \mathbb{E} f(x_j) - \left( \gamma - \frac{(q+n)L\gamma^2}{2n} \right) \mathbb{E} \left\| \nabla f(x_j) \right\|_2^2 + \frac{(1+q)L\gamma^2 \sigma^2}{2mn} + \frac{qL\gamma^2 \varsigma^2}{2n}
\end{aligned} \tag{56}
$$

Summing up from $\mathbb{E} f(x_1)$ to $\mathbb{E} f(x_K)$, we have:

$$\mathbb{E} f(x_K) \leq \mathbb{E} f(x_0) - \left( \gamma - \frac{(q+n)L\gamma^2}{2n} \right) \sum_{j=0}^{K-1} \mathbb{E} \left\| \nabla f(x_j) \right\|_2^2 + \frac{(1+q)L\gamma^2 \sigma^2 K}{2mn} + \frac{qL\gamma^2 \varsigma^2 K}{2n} \tag{57}$$

Thus, we have Theorem 2 if $(1 + \frac{q}{n})L\gamma < 2$:

$$\frac{1}{K} \left( \sum_{j=0}^{K-1} \left\| \nabla f(x_j) \right\|_2^2 \right) \leq \frac{(f(x_0) - f^*)D}{\gamma K} + \frac{(1+q)L\gamma\sigma^2 D}{2M} + \frac{qL\gamma\varsigma^2 D}{2n}, \tag{58}$$

where

$$D = \frac{2}{2 - (1 + \frac{q}{n})L\gamma} \tag{59}$$

$\square$

If we set $\gamma = \theta \sqrt{M/K}$ and let $(1 + \frac{q}{n})L\gamma \leq 1$ (i.e., $D \leq 2$) in Theorem 2, we can obtain Corollary 3.

***Proof to Theorem 3.*** The computation of Algorithm 2 can be expressed as

$$X_{j+1} = \begin{cases} X_j - \gamma \partial F(X_j; \xi_j) & , \text{if } (j+1) \bmod p \neq 0 \\ X_{j+1-p} + Q \left( \left( X_j - \gamma \partial F(X_j; \xi_j) \right) - X_{j+1-p} \right) W_n & , \text{if } (j+1) \bmod p = 0 \end{cases} \tag{60}$$

Here, $X_j = [x_{j,1}, x_{j,2}, ..., x_{j,n}] \in \mathbb{R}^{N \times n}$ are the model parameters on $n$ nodes, $Q$ is any unbiased stochastic quantization function, $W_n$ is a $n$ by $n$ matrix with all elements being $1/n$, and $\partial F(X_j; \xi_j) = [\nabla F_1(x_{j,1}; \xi_{j,1}), \nabla F_2(x_{j,2}; \xi_{j,2}), ..., \nabla F_n(x_{j,n}; \xi_{j,n})]$ are the stochastic gradients.

The above formula can be expressed with projection matrix as

$$X_{j+1} = \mathcal{P}_j(X_j - \widehat{\gamma \partial F(X_j; \xi_j)})W_n + (\mathcal{I} - \mathcal{P}_j)(X_j - \gamma \partial F(X_j; \xi_j)), \tag{61}$$

where $\mathcal{P}_j = \mathcal{I}$ when $j+1$ divides $p$; otherwise $\mathcal{P}_j = \mathbf{0}$, and

$$(X_j - \widehat{\gamma \partial F(X_j; \xi_j)}) = X_{j+1-p} + Q \left( \left( X_j - \gamma \partial F(X_j; \xi_j) \right) - X_{j+1-p} \right) \tag{62}$$

when $(j+1) \bmod p = 0$. Note that $X_j$ have identical columns when $j$ divides $p$.

Consider $\mathbb{E}f\left(X_{j+1}\frac{\mathbf{1}_n}{n}\right)$:

$$\mathbb{E}f\left(X_{j+1}\frac{\mathbf{1}_n}{n}\right) = \mathbb{E}f\left(X_j\frac{\mathbf{1}_n}{n} - \left(\gamma\partial F(X_j;\xi_j)\frac{\mathbf{1}_n}{n} + \mathcal{P}_j\left((X_j - \widehat{\gamma\partial F(X_j;\xi_j)}) - (X_j - \gamma\partial F(X_j;\xi_j))\right)\frac{\mathbf{1}_n}{n}\right)\right)$$

$$=\mathbb{E}f\left(X_j\frac{\mathbf{1}_n}{n} - \left(\gamma\partial F(X_j;\xi_j)\frac{\mathbf{1}_n}{n} + \mathcal{P}_j\left(Q\left((X_j - \gamma\partial F(X_j;\xi_j)) - X_{j+1-p}\right) - (X_j - \gamma\partial F(X_j;\xi_j) - X_{j+1-p})\right)\frac{\mathbf{1}_n}{n}\right)\right)$$

$$\leq\mathbb{E}f\left(X_j\frac{\mathbf{1}_n}{n}\right) - \mathbb{E}\left\langle\nabla f\left(X_j\frac{\mathbf{1}_n}{n}\right), \gamma\partial F(X_j;\xi_j)\frac{\mathbf{1}_n}{n}\right\rangle$$

$$- \mathbb{E}\left\langle\nabla f\left(X_j\frac{\mathbf{1}_n}{n}\right), \mathcal{P}_j\left(Q\left((X_j - \gamma\partial F(X_j;\xi_j)) - X_{j+1-p}\right) - (X_j - \gamma\partial F(X_j;\xi_j) - X_{j+1-p})\right)\frac{\mathbf{1}_n}{n}\right\rangle$$

$$+ \frac{L}{2}\mathbb{E}\left\|\gamma\partial F(X_j;\xi_j)\frac{\mathbf{1}_n}{n} + \mathcal{P}_j\left(Q\left((X_j - \gamma\partial F(X_j;\xi_j)) - X_{j+1-p}\right) - (X_j - \gamma\partial F(X_j;\xi_j) - X_{j+1-p})\right)\frac{\mathbf{1}_n}{n}\right\|_2^2$$

$$=\mathbb{E}f\left(X_j\frac{\mathbf{1}_n}{n}\right) - \underbrace{\mathbb{E}\left\langle\nabla f\left(X_j\frac{\mathbf{1}_n}{n}\right), \gamma\partial f(X_j)\frac{\mathbf{1}_n}{n}\right\rangle}_{=:T_1}$$

$$+ \frac{L}{2}\underbrace{\mathbb{E}\left\|\gamma\partial F(X_j;\xi_j)\frac{\mathbf{1}_n}{n} + \mathcal{P}_j\left(Q\left((X_j - \gamma\partial F(X_j;\xi_j)) - X_{j+1-p}\right) - (X_j - \gamma\partial F(X_j;\xi_j) - X_{j+1-p})\right)\frac{\mathbf{1}_n}{n}\right\|_2^2}_{=:T_2} \tag{63}$$

For $T_1$ we have:

$$T_1 = \frac{\gamma}{2}\mathbb{E}\left\|\nabla f\left(X_j\frac{\mathbf{1}_n}{n}\right)\right\|_2^2 + \frac{\gamma}{2}\mathbb{E}\left\|\partial f(X_j)\frac{\mathbf{1}_n}{n}\right\|_2^2 - \frac{\gamma}{2}\underbrace{\mathbb{E}\left\|\nabla f\left(X_j\frac{\mathbf{1}_n}{n}\right) - \partial f(X_j)\frac{\mathbf{1}_n}{n}\right\|_2^2}_{=:T_3} \tag{64}$$

$$T_3 \leq \frac{L^2}{n}\sum_{i=1}^n\underbrace{\mathbb{E}\left\|X_j\frac{\mathbf{1}_n}{n} - X_j\mathbf{e}_i\right\|_2^2}_{=:T_{4j}} \tag{65}$$

Following the same steps from (32) to (39), we can get the same bound for $\sum_{j=0}^{K-1}T_{4j}$:

$$\sum_{j=0}^{K-1}T_{4j} \leq \frac{pn^2\gamma^2\left(K\left(\frac{\sigma^2}{m} + 3p\varsigma^2\right) + 3p\sum_{j=0}^{K-1}\mathbb{E}\left\|\nabla f\left(X_j\frac{\mathbf{1}_n}{n}\right)\right\|_2^2\right)}{1 - 3np^2L^2\gamma^2} \tag{66}$$

For $T_2$, we have

$$T_2 :=\mathbb{E}\left\|\gamma\partial F(X_j;\xi_j)\frac{\mathbf{1}_n}{n} + \mathcal{P}_j\left(Q\left((X_j - \gamma\partial F(X_j;\xi_j)) - X_{j+1-p}\right) - (X_j - \gamma\partial F(X_j;\xi_j) - X_{j+1-p})\right)\frac{\mathbf{1}_n}{n}\right\|_2^2$$

$$=\mathbb{E}\left\|\gamma\partial F(X_j;\xi_j)\frac{\mathbf{1}_n}{n}\right\|_2^2$$

$$+ \mathbb{E}\left\|\mathcal{P}_j\left(Q\left((X_j - \gamma\partial F(X_j;\xi_j)) - X_{j+1-p}\right) - (X_j - \gamma\partial F(X_j;\xi_j) - X_{j+1-p})\right)\frac{\mathbf{1}_n}{n}\right\|_2^2$$

$$+ \mathbb{E}\left\langle\gamma\partial F(X_j;\xi_j)\frac{\mathbf{1}_n}{n}, \mathcal{P}_j\left(Q\left((X_j - \gamma\partial F(X_j;\xi_j)) - X_{j+1-p}\right) - (X_j - \gamma\partial F(X_j;\xi_j) - X_{j+1-p})\right)\frac{\mathbf{1}_n}{n}\right\rangle$$

$$=\mathbb{E}\left\|\gamma\partial F(X_j;\xi_j)\frac{\mathbf{1}_n}{n}\right\|_2^2$$

$$+\underbrace{\mathbb{E}\left\|\mathcal{P}_j\left(Q\left((X_j-\gamma\partial F(X_j;\xi_j))-X_{j+1-p}\right)-\left(X_j-\gamma\partial F(X_j;\xi_j)-X_{j+1-p}\right)\right)\frac{\mathbf{1}_n}{n}\right\|_2^2}_{=:T_{5j}}$$

$$\overset{\text{According to (40)}}{\leq}\frac{\gamma^2\sigma^2}{mn}+\gamma^2\mathbb{E}\left\|\partial f(X_j)\frac{\mathbf{1}_n}{n}\right\|_2^2+T_{5j}\tag{67}$$

Because $\mathcal{P}_j=\mathbf{0}$ when $(j+1)\bmod p\neq0$, we have $T_{5j}=0$ when $(j+1)\bmod p\neq0$.

When $(j+1)\bmod p=0$,

$$T_{5j}=\mathbb{E}\left\|\left(Q\left((X_j-\gamma\partial F(X_j;\xi_j))-X_{j+1-p}\right)-\left(X_j-\gamma\partial F(X_j;\xi_j)-X_{j+1-p}\right)\right)\frac{\mathbf{1}_n}{n}\right\|_2^2$$

$$=\gamma^2\mathbb{E}\left\|\left(Q\left(\sum_{k=j+1-p}^{j}\partial F(X_k;\xi_k)\right)-\sum_{k=j+1-p}^{j}\partial F(X_k;\xi_k)\right)\frac{\mathbf{1}_n}{n}\right\|_2^2$$

$$=\gamma^2\mathbb{E}\left\|\frac{1}{n}\sum_{i=1}^{n}\left(Q\left(\sum_{k=j+1-p}^{j}\nabla F_i(x_{k,i};\xi_{k,i})\right)-\sum_{k=j+1-p}^{j}\nabla F_i(X_{k,i};\xi_{k,i})\right)\right\|_2^2$$

$$=\frac{\gamma^2}{n^2}\sum_{i=1}^{n}\mathbb{E}\left\|Q\left(\sum_{k=j+1-p}^{j}\nabla F_i(x_{k,i};\xi_{k,i})\right)-\sum_{k=j+1-p}^{j}\nabla F_i(x_{k,i};\xi_{k,i})\right\|_2^2$$

$$\leq\frac{q\gamma^2}{n^2}\sum_{i=1}^{n}\mathbb{E}\left\|\sum_{k=j+1-p}^{j}\nabla F_i(x_{k,i};\xi_{k,i})\right\|_2^2$$

$$=\frac{q\gamma^2}{n^2}\sum_{i=1}^{n}\mathbb{E}\left\|\sum_{k=j+1-p}^{j}\left(\nabla F_i(x_{k,i};\xi_{k,i})-\nabla f_i(x_{k,i})+\nabla f_i(x_{k,i})\right)\right\|_2^2$$

$$=\frac{q\gamma^2}{n^2}\sum_{i=1}^{n}\mathbb{E}\left\|\sum_{k=j+1-p}^{j}\left(\nabla F_i(x_{k,i};\xi_{k,i})-\nabla f_i(x_{k,i})\right)\right\|_2^2+\frac{q\gamma^2}{n^2}\sum_{i=1}^{n}\mathbb{E}\left\|\sum_{k=j+1-p}^{j}\nabla f_i(x_{k,i})\right\|_2^2$$

$$=\frac{q\gamma^2}{n^2}\sum_{i=1}^{n}\sum_{k=j+1-p}^{j}\mathbb{E}\left\|\nabla F_i(x_{k,i};\xi_{k,i})-\nabla f_i(x_{k,i})\right\|_2^2+\frac{q\gamma^2}{n^2}\sum_{i=1}^{n}\mathbb{E}\left\|\sum_{k=j+1-p}^{j}\nabla f_i(x_{k,i})\right\|_2^2$$

$$=\frac{q\gamma^2}{n^2}\sum_{i=1}^{n}\sum_{k=j+1-p}^{j}\mathbb{E}\left\|\frac{1}{m}\sum_{s=1}^{m}\nabla F_i(x_{k,i};\xi_{k,i,s})-\nabla f_i(x_{k,i})\right\|_2^2+\frac{q\gamma^2}{n^2}\sum_{i=1}^{n}\mathbb{E}\left\|\sum_{k=j+1-p}^{j}\nabla f_i(x_{k,i})\right\|_2^2$$

$$\leq\frac{qp\gamma^2\sigma^2}{mn}+\frac{q\gamma^2}{n^2}\sum_{i=1}^{n}\mathbb{E}\left\|\sum_{k=j+1-p}^{j}\nabla f_i(x_{k,i})\right\|_2^2$$

$$\leq\frac{qp\gamma^2\sigma^2}{mn}+\frac{qp\gamma^2}{n^2}\sum_{i=1}^{n}\sum_{k=j+1-p}^{j}\mathbb{E}\left\|\nabla f_i(x_{k,i})\right\|_2^2$$

$$\leq\frac{qp\gamma^2\sigma^2}{mn}+\frac{qp\gamma^2}{n^2}\sum_{i=1}^{n}\sum_{k=j+1-p}^{j}\mathbb{E}\left\|\nabla f_i(x_{k,i})-\nabla f(x_{k,i})+\nabla f(x_{k,i})-\nabla f(X_k\frac{\mathbf{1}_n}{n})+\nabla f(X_k\frac{\mathbf{1}_n}{n})\right\|_2^2$$

$$\leq\frac{qp\gamma^2\sigma^2}{mn}+\frac{3qp^2\gamma^2\varsigma^2}{n}+\frac{3qp\gamma^2}{n^2}\sum_{k=j+1-p}^{j}T_{4k}+\frac{3qp\gamma^2}{n}\sum_{k=j+1-p}^{j}\mathbb{E}\left\|\nabla f(X_k\frac{\mathbf{1}_n}{n})\right\|_2^2$$

$$\tag{68}$$

Plugging $T_1, T_2$ into (63), we can obtain:

$$\mathbb{E}f\left(X_{j+1}\frac{\mathbf{1}_n}{n}\right) \leq \mathbb{E}f\left(X_j\frac{\mathbf{1}_n}{n}\right) - \frac{\gamma}{2}\mathbb{E}\left\|\nabla f\left(X_j\frac{\mathbf{1}_n}{n}\right)\right\|_2^2 - \frac{\gamma}{2}\mathbb{E}\left\|\partial F(X_j)\frac{\mathbf{1}_n}{n}\right\|_2^2 + \frac{L^2\gamma}{2n}T_{4j}$$

$$+ \frac{L\gamma^2\sigma^2}{2mn} + \frac{L\gamma^2}{2}\mathbb{E}\left\|\partial f(X_j)\frac{\mathbf{1}_n}{n}\right\|_2^2 + \frac{L}{2}T_{5j} \tag{69}$$

Suppose

$$\frac{\gamma}{2} > \frac{L\gamma^2}{2}, \tag{70}$$

we have

$$\mathbb{E}f\left(X_{j+1}\frac{\mathbf{1}_n}{n}\right) \leq \mathbb{E}f\left(X_j\frac{\mathbf{1}_n}{n}\right) - \frac{\gamma}{2}\mathbb{E}\left\|\nabla f\left(X_j\frac{\mathbf{1}_n}{n}\right)\right\|_2^2 + \frac{L^2\gamma}{2n}T_{4j}$$

$$+ \frac{L\gamma^2\sigma^2}{2mn} + \frac{L}{2}T_{5j} \tag{71}$$

Summing up from $\mathbb{E}f(x_1)$ to $\mathbb{E}f(x_K)$, because $T_{5j} = 0$ when $(j+1) \bmod p \neq 0$, we can obtain:

$$\mathbb{E}f\left(X_K\frac{\mathbf{1}_n}{n}\right) \leq \mathbb{E}f\left(x_0\right) - \frac{\gamma}{2}\sum_{j=0}^{K-1}\mathbb{E}\left\|\nabla f\left(X_j\frac{\mathbf{1}_n}{n}\right)\right\|_2^2 + \frac{L^2\gamma}{2n}\sum_{j=0}^{K-1}T_{4j}$$

$$+ \frac{L\gamma^2\sigma^2 K}{2mn} + \frac{L}{2}\left(\frac{q\gamma^2\sigma^2 K}{mn} + \frac{3qp\gamma^2\varsigma^2 K}{n} + \frac{3qp\gamma^2}{n^2}\sum_{j=0}^{K-1}T_{4j} + \frac{3qp\gamma^2}{n}\sum_{j=0}^{K-1}\mathbb{E}\left\|\nabla f(X_k\frac{\mathbf{1}_n}{n})\right\|_2^2\right)$$

$$= \mathbb{E}f\left(x_0\right) - \left(\frac{\gamma}{2} - \frac{3Lqp\gamma^2}{2n}\right)\sum_{j=0}^{K-1}\mathbb{E}\left\|\nabla f\left(X_j\frac{\mathbf{1}_n}{n}\right)\right\|_2^2$$

$$+ \left(\frac{L^2\gamma}{2n} + \frac{3Lqp\gamma^2}{2n^2}\right)\sum_{j=0}^{K-1}T_{4j} + \frac{(1+q)L\gamma^2\sigma^2 K}{2mn} + \frac{3Lqp\gamma^2\varsigma^2 K}{2n}$$

$$\tag{72}$$

Plugging (66) into (72), if $1 - \frac{3Lqp\gamma}{n} - \frac{3p^2\gamma^2(nL^2+3qpL\gamma)}{1-3np^2L^2\gamma^2} > 0$, we can obtain:

$$\frac{1}{K}\sum_{j=0}^{K-1}\mathbb{E}\left\|\nabla f\left(X_j\frac{\mathbf{1}_n}{n}\right)\right\|_2^2 \leq \frac{2\left(f(x_0) - f^*\right)D}{\gamma K} + \frac{(1+q)L\gamma\sigma^2 D}{M} + \frac{3Lqp\gamma\varsigma^2 D}{n}$$

$$+ \frac{(nL^2+3qpL\gamma)p\gamma^2\sigma^2 D}{m(1-3np^2L^2\gamma^2)} + \frac{3(nL^2+3qpL\gamma)p^2\gamma^2\varsigma^2 D}{1-3np^2L^2\gamma^2} \tag{73}$$

where

$$D = \frac{1}{\left(1 - \frac{3Lqp\gamma}{n} - \frac{3p^2\gamma^2(nL^2+3qpL\gamma)}{1-3np^2L^2\gamma^2}\right)} \tag{74}$$

If we set $\gamma = \theta\sqrt{M/K}$ and let

$$\frac{3Lqp\gamma}{n} \leq \frac{1}{4} \tag{75}$$

and

$$\frac{3p^2\gamma^2(nL^2+3qpL\gamma)}{1-3np^2L^2\gamma^2} \leq \frac{3Lqp\gamma}{n}, \tag{76}$$

we can get $D \leq 2$. Thus,

$$\frac{1}{K}\sum_{j=0}^{K-1}\mathbb{E}\left\|\nabla f\left(X_j\frac{\mathbf{1}_n}{n}\right)\right\|_2^2 \leq \frac{2\theta^{-1}\left(f(x_0) - f^*\right)}{\sqrt{MK}} + \frac{2(1+q)L\theta\sigma^2}{\sqrt{MK}} + \frac{6qpmL\theta\varsigma^2}{\sqrt{MK}}$$

$$+ \frac{2qL\gamma\sigma^2}{mn} + \frac{6qpL\gamma\varsigma^2}{n}$$

$$= \frac{2\theta^{-1}\left(f(x_0) - f^*\right)}{\sqrt{MK}} + \frac{2(1+q)L\theta\sigma^2}{\sqrt{MK}} + \frac{6qpmL\theta\varsigma^2}{\sqrt{MK}}$$

$$+ \frac{2qL\theta\sigma^2}{\sqrt{MK}} + \frac{6qpmL\theta\varsigma^2}{\sqrt{MK}}$$

$$= \frac{2\theta^{-1}\left(f(x_0) - f^*\right)}{\sqrt{MK}} + \frac{2(1 + 2q)L\theta\sigma^2}{\sqrt{MK}} + \frac{12qpmL\theta\varsigma^2}{\sqrt{MK}} \tag{77}$$

The condition in (76) is equivalent to

$$\gamma = \theta\sqrt{\frac{M}{K}} \leq \frac{\sqrt{(n^2pL)^2 + 12np^2q^2(1 + L^2)} - n^2pL}{6np^2q(1 + L^2)} \tag{78}$$

which simplifies to

$$K \geq \frac{M\theta^2\left(\sqrt{(n^2pL)^2 + 12np^2q^2(1 + L^2)} + n^2pL\right)^2}{4q^2} \tag{79}$$

We can see that if $K$ is large enough, the conditions in (79), (75) and (70) can be satisfied and thus the bound in (77) is valid, which gives us Theorem 3.

$\square$

# B  Error bound of QSGD proposed by Alistarh *et al.* [5]

Alistarh *et al.* [5] propose a quantization function controlled by a quantization level $s$. A gradient component $v_i$ is quantized to either $l/s$ or $(l+1)/s$ based on a Bernoulli distribution defined as

$$\begin{cases} \mathbb{P}\{b_i = (l+1)/s\} = \frac{s|v_i|}{\|v\|_2} - l \\ \mathbb{P}\{b_i = l/s\} = 1 - \frac{s|v_i|}{\|v\|_2} + l \end{cases}$$

The quantized value of $v_i$ is defined as $Q_s(v_i) = b_i \cdot sign(v_i) \cdot \|v\|_2$.

Because

$$\mathbb{E}\left[Q_s(v_t)\right] = \frac{l+1}{s} \cdot sign(v_i) \cdot \|v\|_2 \cdot \left(\frac{s|v_i|}{\|v\|_2} - l\right) + \frac{l}{s} \cdot sign(v_i) \cdot \|v\|_2 \cdot \left(1 - \frac{s|v_i|}{\|v\|_2} + l\right) = v_i,$$

$Q_s$ is unbiased.

Alistarh *et al.* have shown the error bound $q$ for $Q_s$ is $\min(N/s^2, \sqrt{N}/s)$ [5], which suggest $s$ should be set to $\sqrt{N}$ to achieve $q = 1$.

We now show a tighter bound of $q$ is $\min(N/(4s^2), \sqrt{N}/s)$. Because

$$\left\|Q_s(v_i) - v_i\right\|_2^2 = \left(\frac{l+1}{s} \cdot \|v\|_2 - |v_i|\right)^2$$

with probability $\frac{s|v_i|}{\|v\|_2} - l$, and

$$\left\|Q_s(v_i) - v_i\right\|_2^2 = \left(\frac{l}{s} \cdot \|v\|_2 - |v_i|\right)^2$$

with probability $1 - \frac{s|v_i|}{\|v\|_2} + l$, we have

$$\mathbb{E}\left\|Q_s(v_i) - v_i\right\|_2^2 = \left(\frac{l+1}{s} \cdot \|v\|_2 - |v_i|\right)^2 \left(\frac{s|v_i|}{\|v\|_2} - l\right) + \left(\frac{l}{s} \cdot \|v\|_2 - |v_i|\right)^2 \left(1 - \frac{s|v_i|}{\|v\|_2} + l\right)$$

$$= \left(\left(\frac{l+1}{s} \cdot \|v\|_2 - |v_i|\right)^2 - \left(\frac{l}{s} \cdot \|v\|_2 - |v_i|\right)^2\right)\left(\frac{s|v_i|}{\|v\|_2} - l\right)$$

$$+ \left(\frac{l}{s} \cdot \|v\|_2 - |v_i|\right)^2$$

$$= \left(\frac{\|v\|_2}{s}\left(\frac{2l+1}{s} \cdot \|v\|_2 - 2|v_i|\right)\right)\left(\frac{s|v_i|}{\|v\|_2} - l\right) + \left(\frac{l}{s} \cdot \|v\|_2 - |v_i|\right)^2$$

$$= \frac{(2l+1)|v_i|\|v\|_2}{s} - v_i^2 - \frac{l^2+l}{s^2}\|v\|_2^2$$

$$= \left(|v_i| - \frac{l\|v\|_2}{s}\right)\left(\frac{(l+1)\|v\|_2}{s} - |v_i|\right) \tag{80}$$

$$\overset{\because |v_i| \in \left[\frac{l\|v\|_2}{s}, \frac{(l+1)\|v\|_2}{s}\right)}{\leq} \frac{\|v\|_2^2}{4s^2} \tag{81}$$

Therefore,

$$\mathbb{E}\left\|Q_s(v) - v\right\|_2^2 \leq \frac{N\|v\|_2^2}{4s^2} \tag{82}$$

It follows that

$$\frac{\mathbb{E}\left\|Q_s(v) - v\right\|_2^2}{\|v\|_2^2} \leq \frac{N}{4s^2} \tag{83}$$

From (80) we can also get:

$$\mathbb{E}\left\|Q_s(v_i) - v_i\right\|_2^2 = \left(|v_i| - \frac{l\|v\|_2}{s}\right)\left(\frac{(l+1)\|v\|_2}{s} - |v_i|\right) \overset{\because |v_i| \in \left[\frac{l\|v\|_2}{s}, \frac{(l+1)\|v\|_2}{s}\right)}{\leq} \left(|v_i| - \frac{l\|v\|_2}{s}\right)\left(\frac{\|v\|_2}{s}\right)$$

$$= \left(1 - \frac{l\|v\|_2}{s|v_i|}\right)\left(\frac{\|v\|_2|v_i|}{s}\right) \leq \frac{\|v\|_2|v_i|}{s} \tag{84}$$

Thus,

$$\mathbb{E}\big\|Q_s(v) - v\big\|_2^2 \leq \frac{\|v\|_2\|v\|_1}{s} \overset{\because \|v\|_1 \leq \sqrt{N}\|v\|_2}{\leq} \frac{\sqrt{N}\|v\|_2^2}{s} \tag{85}$$

It follows that

$$\frac{\mathbb{E}\big\|Q_s(v) - v\big\|_2^2}{\|v\|_2^2} \leq \frac{\sqrt{N}}{s} \tag{86}$$

Combining (83) and (86), we have

$$\frac{\mathbb{E}\big\|Q_s(v) - v\big\|_2^2}{\|v\|_2^2} \leq \min\left(\frac{N}{4s^2}, \frac{\sqrt{N}}{s}\right) \tag{87}$$

This result suggest that $s = \frac{\sqrt{N}}{2}$ can achieve $q = 1$.

However, this number is not the actual quantization level of this function, because $l$ has a smaller bound $\frac{\sqrt{N}\|v\|_\infty}{2\|v\|_2}$. To illustrate this point, we present an equivalent quantization function $Q_a$ as follows:

For a gradient vector $v$, we first identify $\|v\|_\infty$ and divide it into $\frac{\sqrt{N}\|v\|_\infty}{2\|v\|_2}$ intervals of size $\phi = \frac{2\|v\|_2}{\sqrt{N}}$. The absolute value of other components will fall into one of the intervals. Suppose a component $v_i$ is in interval $[l\phi, (l+1)\phi)$, its quantized value is either the upper bound or the lower bound based on a Bernoulli distribution defined as

$$\begin{cases} \mathbb{P}\{b_i = (l+1)\} = \frac{|v_i|}{\phi} - l \\ \mathbb{P}\{b_i = l\} = 1 - \frac{|v_i|}{\phi} + l \end{cases} \tag{88}$$

The quantized value of $v_i$ is defined as $Q_a(v_i) = b_i \cdot sign(v_i) \cdot \phi$. Because

$$\big\|Q_a(v_i) - v_i\big\|_2^2 = \begin{cases} (|v_i| - l\phi)^2 & \text{, if } b_i = l \\ (|v_i| - (l+1)\phi)^2 & \text{, if } b_i = (l+1) \end{cases}$$

we have

$$\mathbb{E}\big\|Q_a(v_i) - v_i\big\|_2^2 = (|v_i| - l\phi)\phi - (|v_i| - l\phi)^2 \leq \frac{\phi^2}{4}, \text{ for } v_i \in [l\phi, (l+1)\phi)$$

which implies $\mathbb{E}\big\|Q_a(v) - v\big\|_2^2 \leq \|v\|_2^2, \forall v \in \mathbb{R}^N$. Thus, $Q_a$ ensures $q = 1$. It is not hard to see that $Q_a$ is unbiased.

$Q_a$ is actually equivalent to $Q_s$ proposed by Alistarh *et al.* because both function use interval of size $\phi = \frac{2\|v\|_2}{\sqrt{N}}$. The presentation of $Q_a$ shows that the number of intervals is actually $\frac{\sqrt{N}\|v\|_\infty}{2\|v\|_2}$, which means that the number of bits to represent the quantized value is $\left\lceil \log_2\left(\frac{\sqrt{N}\|v\|_\infty}{2\|v\|_2}\right) \right\rceil$.

# C Error bound of TernGrad proposed by Wen *et al.* [46]

For a gradient vector $v$, they first identify the component with the largest absolute value (i.e., $\|v\|_\infty$). Then, the gradient components are quantized to either 1 or 0 based on a Bernoulli distribution defined as

$$\begin{cases} \mathbb{P}\{b_i = 1\} = |v_i| / \|v\|_\infty \\ \mathbb{P}\{b_i = 0\} = 1 - |v_i| / \|v\|_\infty \end{cases}$$

The quantized value of $v_i$ is defined as $Q_t(v_i) = b_i \cdot sign(v_i) \cdot \|v\|_\infty$. Because $\mathbb{E}\left[Q_t(v_i)\right] = sign(v_i) \cdot \|v\|_\infty \cdot |v_i| / \|v\|_\infty + 0 \cdot \left(1 - |v_i| / \|v\|_\infty\right) = v_i$, $Q_t$ is unbiased.

We now consider its error bound. Because

$$\left\|Q_t(v_i) - v_i\right\|_2^2 = \begin{cases} v_i^2 & \text{, if } b_i = 0 \\ (\|v\|_\infty - |v_i|)^2 & \text{, if } b_i = 1 \end{cases}$$

we have

$$\mathbb{E}\left\|Q_t(v_i) - v_i\right\|_2^2 = \|v\|_\infty |v_i| - v_i^2$$

and thus,

$$\mathbb{E}\left\|Q_t(v) - v\right\|_2^2 = \|v\|_\infty \|v\|_1 - \|v\|_2^2$$

Since $\|v\|_\infty \le \|v\|_2 \le \|v\|_1 \le \sqrt{N}\|v\|_2, \forall v \in \mathbb{R}^N$, we have $\mathbb{E}\left\|Q_t(v) - v\right\|_2^2 \le (\sqrt{N} - 1)\|v\|_2^2$. Therefore, $q = \sqrt{N} - 1$, and `TernGrad` cannot achieve $O(1/\sqrt{MK})$ convergence rate in general.

# D  More experimental results on image classification

(a) VGG19 on 4 machines ($m = 32$, $\gamma = 0.1$ for epoch $0 - 109$, $\gamma = 0.01$ for epoch $110 - 159$ )

(b) ResNet50 on 8 machines ($m = 32$, $\gamma = 0.2$ for epoch $0 - 109$, $\gamma = 0.02$ for epoch $110 - 199$ )

(c) ResNet110 on 8 machines ($m = 32$, $\gamma = 0.01$ for epoch $0 - 9$, $\gamma = 0.2$ for epoch $10 - 109$, $\gamma = 0.02$ for epoch $110 - 199$ )

(d) ResNet50 on 8 machines ($m = 32$, $\gamma = 0.2$ for epoch $0 - 109$, $\gamma = 0.02$ for epoch $110 - 199$ )

Figure 2: More results on image classification

Figure 2a and 2b validate that sparse parameter averaging can achieve $\mathcal{O}(1/\sqrt{MK})$ convergence rate. Figure 2c and 2d validate that 1) QSGD can achieve $\mathcal{O}(1/\sqrt{MK})$ convergence rate if configured properly, 2) our PQASGD can achieve $\mathcal{O}(1/\sqrt{MK})$ convergence rate with even less communication overhead, and 3) partitioned training data does have a negative effect to the convergence rate of distributed SGD with quantized gradients.

# E  Results on speech recognition

Figure 3: WER and training loss for a 5-layer LSTM on AN4.

We train a 5-layer LSTM of 800 hidden units per layer on AN4 dataset. The total mini-batch size $M$ is set to 128. The learning rate is initialized to $3e^{-4}$ and decays in each epoch with annealing rate 1.01 (the learning rate is constant over iterations within an epoch).

Figure 3a compares the word error rate of FULLSGD with PSGD and RSGD with different settings of $p$ on partitioned training data. We can see that, as the learning rate decreases, PSGD and RSGD match FULLSGD after 50 epochs. This validates our claim in Theorem 1 that sparse parameter averaging can achieve $\mathcal{O}(1/\sqrt{MK})$ convergence rate if $K$ is large enough.

Figure 3b compares the word error rate of FULLSGD with different versions of QSGD and our PQASGD. We can observe a clear gap on the zoomed figure between FULLSGD and QSGD-P with $s$ set to $\sqrt{N}/2$. This validates our claim that data partitioning affects the convergence rate of distributed SGD with gradient quantization. This effect is eliminated by setting $s$ to $\sqrt{mN}/2$ for QSGD-P as we can see that all the other versions have similar WER after 50 epochs. Though our PQASGD converges slower than other versions in the first 40 epochs, it matches the WER of FULLSGD after 50 epochs. This validates our claim that our PQASGD converges at rate $\mathcal{O}(1/\sqrt{MK})$. Our PQASGD requires only 4.5% communication data size compared with FULLSGD in this case.

We then remove the validation at the end of each epoch and measure the execution time for training. Figure 3c shows the training loss over time for FULLSGD, QSGD and PQASGD on partitioned training data. We can see that our PQASGD runs 1.4x faster than QSGD and 2.2x faster than FULLSGD.

As shown in Figure 3d to 3f, the results on 16 GPUs follow a similar pattern.