[Reviews · NeurIPS 2018]

Reviewer 1



======== after rebuttal Some of the concerns I had about the paper have been addressed by the authors' response (eg in relation to convergence proofs), yet some still stand. Especially the ones related to experiments, and parameter tuning. Since the authors appropriately responded to some of my key criticisms I will upgrade my score from 5 to 6. ========original review This paper analyzes the convergence rate of distributed mini-batch SGD using sparse and quantized communication with application to deep learning. Based on the analysis, it proposes combining sparse and quantized communication to further reduce the communication cost that burdens the wall clock runtime in distributed setups. The paper is generally well written. The convergence analysis does appear to make sense, and the proposed combination of sparsification and quantization seems to save runtime a little bit with proper parameter tuning. However, I have a few concerns about this work, listed below: - A concern about this work is the novelty of the analysis. The key idea of proving convergence with noisy gradient update is bounding the variance of the gradient while assuming the noisy gradient update is unbiased, which, has been known for a long time, eg see [1-3] below. In fact, the following two papers, although not discussed in the submitted paper, have given detailed ways of minimizing variance for sparsified+quantized gradients. [1] Jakub Konečný and Peter Richtárik, Randomized Distributed Mean Estimation: Accuracy vs Communication, 2016. [2] Ananda Theertha Suresh et al, Distributed Mean Estimation with Limited Communication, 2017. [3] Jianqiao Wangni et al. Gradient sparsification for communication-efficient distributed optimization. arXiv preprint arXiv:1710.09854, 2017. Given the above work, it is unclear what are the new insights provided by this paper's theoretical analysis. One can simply prove bounds on the Variance of the quant./sparse gradients (assuming they are unbiased), and a convergence rate can be shown in a plug an play way. As such, it is important to compare the variance of the proposed techniques, for a given communication budget, in comparison to prior art, as listed above. - Furthermore, it is not clear why the proposed technique, i.e., using a combination of sparsification and quantization, will be significantly better than standalone sparsification or quantization. As the theoretical convergence bound in the paper (line 255) suggests, the proposed approach converge at rate O(1+q/ sqrt(MK) ), which is worse than that using sparse communication (O(1/sqrt(MK) ), line 173) and that of using quantized communication (O(1/sqrt(MK) ), line 211). The authors do argue that when the parameters are well chosen, the rate can be roughly the same, but this does not explain why a combination can provide an improvement. I might have missed something on that part of the text. - A key challenge in understanding the performance of distributed SGD using sparse and quantized Communication is how the speedup changes as the compression ratio (either quant. or sparsification) changes. That is for epsilon accuracy, how faster does training happen on P machines versus 1, and what is the best selection of sparsity/quantization levels? Unfortunately, there is no detailed analysis or any experiment shown in the main paper about this. It is critical to have such results (at least an empirical study) to understand the convergence rate in terms of gradient compression efficiency. - In their experimental section, the authors compare the proposed technique with QSGD, terngrad, and vanilla sgd on a specific set of parameters. Where the parameters all tuned for all methods? Eg were the learning rates optimized individually for both QSGD and terngrad, and all setups of compute nodes? It is not clear if the speedup will be significant with well-chosen parameters for QSGD and terngrad, and this is necessary for a fair comparison with prior art. I overall think that the ideas presented in this paper are useful and interesting, but some work still needs to be done in order to clarify some points of confusion. My most major concern is how the proposed algorithm performs experimentally when all other algorithms (eg QSGD) operate with tuned hyperparameters. Finally, since the paper claims to establish linear speedups, these should be also proved experimentally in comparison to prior art.

Reviewer 2



The authors study the convergence of distributed, synchronous, stochastic gradient descent for Lipschitz continuous (possibly non-convex) loss functions for two settings: 1) only a subset of the components gradients are exchanged in each iteration, and 2) the gradients are quantized at each node before being exchanged. These variants of the full-fledged distributed SGD were proposed previously because they save on communication costs, but it appears that their convergence properties were not analyzed prior to this submission. The authors derive explicit conditions on the quantization mechanism, as well as the components exchanged to ensure that the convergence rate corresponds to full data exchange. The authors insights via their analysis also lead to a new variant that combines the two methods of saving on communication cost. The main body of the paper is well written, and the model and the results are clearly explained. The main negative of the paper is that the proofs in the supplemental material are presented without any structure as a long and tedious chain of inequalities. This makes the proof difficult to parse and verify. The results seem significant and important to the NIPS community. I request one clarification: Empirically the authors show that the communication efficient variants converge to the true gradient. However, it is not clear from the simulations if these variants would indeed save on communication if the error is fixed, since they need more number of iterations compared with the full-communication counterpart. It will be good if the authors can compare in future versions of the paper through simulations, the total communication for a fixed error threshold, of various algorithms. ====== I have read the other reviews and the author response. Although there are still some concerns relating providing experimental details (specifically tuning the learning rate for comparable algorithms), the main contributions/novelty of the paper stands in light of the responses. So I stick with my original score.

Reviewer 3



In the paper, the author provides the proof of stochastic gradient sparsification and quantization for non-convex problem and proposes a combined method to reduce the communication time. Sparse gradient [1] [2] and quantized gradient [3][4] have been explored before, however, the deep understanding of its convergence rate for non-convex problem is missing. The author fills an important gap between practical application and theoretical analysis. Periodic quantized averaging SGD is a straightforward extension of sparse gradient and quantized gradient. Experimental results also verify that the proposed method can obtain better speedup. The following are my concerns: 1) the sparse parameter average is the same as communication-efficient distributed methods. There are missing citations about these methods, for example elastic averaging sgd[5]. 2) according to [5], simply averaging the parameters in a period may lead to divergence. Did you meet this problem? Can you report the accuracies of the compared methods? [1] A. F. Aji and K. Heafield. Sparse communication for distributed gradient descent. CoRR, abs/1704.05021, 2017 [2] Lin, Yujun, et al. "Deep gradient compression: Reducing the communication bandwidth for distributed training." arXiv preprint arXiv:1712.01887 (2017) [3]F. Seide, H. Fu, J. Droppo, G. Li, and D. Yu. 1-bit stochastic gradient descent and application to data-parallel distributed training of speech dnns, September 2014 [4] W. Wen, C. Xu, F. Yan, C. Wu, Y. Wang, Y. Chen, and H. Li. Terngrad: Ternary gradients to reduce communication in distributed deep learning. In I. Guyon, U. V. Luxburg, S. Bengio, H. Wallach, R. Fergus, S. Vishwanathan, and R. Garnett, editors, Advances in Neural Information Processing Systems 30, pages 1509–1519. Curran Associates, Inc., 2017 [5] Zhang, Sixin, Anna E. Choromanska, and Yann LeCun. "Deep learning with elastic averaging SGD." Advances in Neural Information Processing Systems. 2015 ======== after rebuttal====== My concerns are addressed. I will update my score to accept.

Reviewer 4



summary: This paper provides theoretical convergence analysis on two recent methods that reduces communication cost in distributed learning, i.e., sparse parameter averaging and gradient quantization. This paper also provides a new method along with theoretical convergence analysis that combines these two strategies. Experiments are performed on CIFAR-10 image classification and speech recognition. Quality: The paper is well organized, and combination of these two strategies are well motivated. The overall technical content of the paper appears to be correct. There are many notations involved in the paper, making some parts a bit hard to follow. There is certainly room for improvements in the clarity of the paper. Some concerns about the theoretical results are: - The scaling factor of the O(1/\sqrt{MK}) rate may become larger due to sparsification or quantization compared to FULLSGD, is it possible that it exceeds \sqrt{M}? In particular, the scaling is proportional to the quantization error q which may be related to the data dimension N (line 219), and may be much larger than \sqrt{M}. In this case, the communication cost reduced by distributed learning cannot even compensate for the slowed convergence due to sparsificaiton and quantization. - In line 220, the quantization level s is proportional to the square root of the data dimension N, which can be very large empirically. In this case, the number of bits used to encode the gradients can also be large. Does this indicate that the O(1/\sqrt{MK}) rate only works for gradient quantization with large number of bits? How many bits are required to guarantee the convergence rate in the paper? How many bits are used empirically? Some points about the experiments: - Currently, there exists much wider bandwidth networks than 1Gbps, what is the performance of the proposed method using network with larger bandwidth? - Besides CIFAR-10, what about the results on larger datasets like imagenet? Will the proposed method still have similar performance as FULLSGD and the other compared methods? - What is the exact number of bits used for QSGD in the experiments? Does QSGD also have requirements on the number of bits for their theory? Will the proposed method still have improvement when smaller number of bits is used for QSGD?